# CHROMA: Consistent HaRmonization Of Multi-View Appearance via Bilateral Grid Prediction

**Jisu Shin**[1,2*]          **Richard Shaw**[1]          **Seunghyun Shin**[2]

**Zhensong Zhang**[1]          **Hae-Gon Jeon**[3†]          **Eduardo Pérez-Pellitero**[1†]

[1] Huawei Noah's Ark Lab
[2] Department of AI Convergence, GIST
[3] Department of Artificial Intelligence, Yonsei University

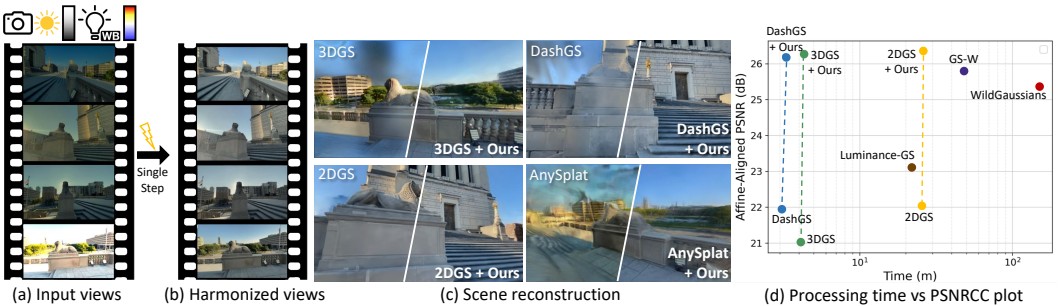

Figure 1: (a) Input views with inconsistent appearance, (b) input views harmonized by our model, (c) novel view renderings of 3DGS fitted to inconsistent input views and ones corrected by our model, (d) comparison with 3DGS-based appearance embedding methods on varying exposure dataset.

## Abstract

Modern camera pipelines apply extensive on-device processing, such as exposure adjustment, white balance, and color correction, which, while beneficial individually, often introduce photometric inconsistencies across views. These appearance variations violate multi-view consistency and degrade novel view synthesis. Joint optimization of scene-specific representations and per-image appearance embeddings has been proposed to address this issue, but with increased computational complexity and slower training. In this work, we propose a generalizable, feed-forward approach that predicts spatially adaptive bilateral grids to correct photometric variations in a multi-view consistent manner. Our model processes hundreds of frames in a single step, enabling efficient large-scale harmonization, and seamlessly integrates into downstream 3D reconstruction models, providing cross-scene generalization without requiring scene-specific retraining. To overcome the lack of paired data, we employ a hybrid self-supervised rendering loss leveraging 3D foundation models, improving generalization to real-world variations. Extensive experiments show that our approach outperforms or matches the reconstruction quality of existing scene-specific optimization methods with appearance modeling, without significantly affecting the training time of baseline 3D models.

## 1 Introduction

Novel view synthesis (NVS) and 3D reconstruction are fundamental challenges in computer vision and graphics. Recent advances, such as Neural Radiance Fields (NeRF) (Mildenhall et al., 2020) and 3D Gaussian Splatting (3DGS) (Kerbl et al., 2023), have significantly improved the fidelity and realism of scene reconstruction and rendering. These methods typically rely on multi-view images

---

*Work done during an internship at Huawei Noah's Ark Lab, UK

†Corresponding authors

captured under the assumption of photometric consistency across views. However, this assumption often breaks in real-world scenarios due to various sources of photometric inconsistency, including: (i) in-camera Image Signal Processing (ISP) variations, e.g. exposure, white balance, color correction; and (ii) scene illumination. Such inconsistencies degrade reconstruction quality, producing floaters, color artifacts, or generally unstable results. To address these challenges, prior art has explored per-view appearance embeddings (Martin-Brualla et al., 2021; Kulhanek et al., 2024; Wang et al., 2024a; Xiao et al., 2025; Cui et al., 2025), providing reconstruction with additional capacity to capture per-view appearance variations via MLPs, tone curves, or affine transforms that are jointly learned with the scene in a global multi-view optimization. While effective, these approaches tightly couple appearance modeling with geometry reconstruction, incurring additional computation at each optimization step, substantially increasing training cost. This undermines the efficiency of pipelines designed for speed, e.g. 3DGS and recent extensions (Mallick et al., 2024; Chen et al., 2025) which achieve rapid scene fitting. Moreover, the recovered appearance after removing the embeddings is not explicitly controllable and often converges to the mean of all variations observed in the input views.

These limitations motivate decoupling appearance harmonization from scene optimization and addressing it in a feed-forward manner, promising improvements in accuracy, controllability, and efficiency. However, several challenges remain when treating this as an independent problem. Existing 2D image and video enhancement techniques (Afifi et al., 2021; Cui et al., 2024; Li et al., 2024; Zhou et al., 2024; Cui et al., 2022; Shin et al., 2023; 2024a) often lack temporal or multi-view consistency, address only limited types of appearance variation (e.g. exposure correction), and struggle to robustly handle severe color shifts or saturation artifacts. Additionally, most approaches require fully-supervised learning paradigms, yet collecting paired real-world data is infeasible: real-world appearance variations are inherently unique in time and space, and one cannot easily capture pixel-aligned images where several appearance variations can be isolated and used as labels.

In this work, we address the aforementioned challenges by introducing a feed-forward approach to multi-view appearance harmonization tailored for 3D reconstruction from varying appearance images. Given multi-view captures of a static scene and a reference frame with a desired appearance, our model transforms all views to match the reference, ensuring photometric consistency. Our key idea is to learn per-frame 3D bilateral grids of affine transforms in a generalizable, multi-view consistent manner. We use bilateral grids as they are compact and expressive, capable of modeling a wide range of ISP operations (Chen et al., 2007; Gharbi et al., 2017; Shin et al., 2024b) and beyond. A multi-view aware transformer predicts low-resolution bilateral grids for each view, which, when applied to the inputs, aligns their appearance with the reference at full resolution. To make the model uncertainty-aware, we also predict bilateral confidence grids using a probabilistic loss. We introduce a reference selection strategy that identifies the image most representative of overall scene appearance, avoiding outliers that could degrade reconstruction quality. Selection is driven by a weighted ranking score accounting for both intensity and semantic similarity, ensuring the final appearance is realistic (i.e. not an average over all frames), while providing flexibility for either manual choice or automatic selection. Lastly, we address the lack of training data for such a system by proposing (i) a synthetic data-generation pipeline to handle large granularity image-level variations, and (ii) a self-supervised training paradigm that uses recent large 3D reconstruction models as a pretext task to provide real-data training signals.

Unlike prior art, our lightweight transformer model introduces only a fixed computational cost per frame. This decoupled design allows seamless integration into pipelines such as 3DGS (Kerbl et al., 2023), 2DGS (Huang et al., 2024a), DashGS (Chen et al., 2025), and even feed-forward models (Jiang et al., 2025), enhancing view consistency while preserving scalability and speed. Through our hybrid self-supervised and supervised (synthetic data) approach, our model achieves state-of-the-art accuracy for real-world datasets, which in turn also improves the robustness and stability of 3DGS optimization under challenging photometric conditions without negatively affecting training time. Comprehensive evaluation demonstrates that our model matches and often exceeds the performance of existing 3DGS-based appearance embedding approaches while maintaining competitive training speed.

## 2 RELATED WORKS

**Appearance Correction and Bilateral Grids.** Image correction aims to adjust visual attributes such as exposure, white balance, and tone to improve image quality or ensure consistency. Traditional

methods solve this using histogram equalization (Zuiderveld, 1994), retinex-based methods, or global transformation optimization, but often lack spatial adaptability. Learning-based approaches (Afifi et al., 2021; Afifi & Brown, 2020; Zhou et al., 2024) address these issues using CNNs, but struggle with generalization or fine detail preservation. Bilateral filtering has been widely used due to its edge-aware properties. Numerous approaches improve its efficiency, such as convolution pyramids (Farbman et al., 2011) and fast bilateral filtering (Paris & Durand, 2006; Tomasi & Manduchi, 1998; Chen et al., 2007). A common acceleration strategy applies the operator at low resolution and upsamples the result, but this results in blurry outputs. Bilateral space optimization (Barron et al., 2015; Barron & Poole, 2016) addresses this by solving an optimization problem within a bilateral grid. Similarly, (Chen et al., 2007) approximate an image operator using a grid of local affine models in bilateral space, where parameters are fit to a single input-output pair. (Gharbi et al., 2017) build upon this by training a neural network to apply the operator to unseen inputs. While most bilateral grid methods operate solely on single 2D images, our work extends this concept to the spatio-temporal domain, enabling multi-view consistent enhancement through a transformer-based architecture.

**Novel View Synthesis under Appearance Variations.** Extensions to NeRF (Mildenhall et al., 2020) and 3DGS (Kerbl et al., 2023) have attempted to solve novel view synthesis under real-world conditions such as inconsistent lighting, occlusions, and scene variability. The pioneering work NeRF-W (Martin-Brualla et al., 2021) incorporates per-image appearance and transient embeddings, with aleatoric uncertainty for transient object removal. Follow-ups improved NeRF robustness (Chen et al., 2021; Yang et al., 2023; Tancik et al., 2022), but suffer from slow optimization, rendering, and limited scalability. In low-light, RAW-NeRF (Mildenhall et al., 2021) leverages raw sensor data, but is constrained by long training times. For 3DGS, VastGaussian (Lin et al., 2024) applies CNNs to 3DGS outputs, but struggles with large appearance shifts. GS-W (Zhang et al., 2024) and WE-GS (Wang et al., 2024b) use CNN-derived reference features, while SWAG (Dahmani et al., 2024) and Scaffold-GS (Lu et al., 2024) store appearance data in a hash-grid-based implicit field (Müller et al., 2022). WildGaussians (Kulhanek et al., 2024) embeds appearance vectors within Gaussians, while Splatfacto-W (Xu et al., 2024) similarly combines Gaussian and image embeddings via an MLP to output spherical harmonics. Luminance-GS (Cui et al., 2025) predicts per-view color transforms followed by view-adaptive curve adjustment. DAVIGS (Lin et al., 2025) learns per-pixel affine transforms using an MLP combining per-view embeddings and 3D features. Most relevant to ours is BilaRF (Wang et al., 2024a), a NeRF-based method learning per-view bilateral grids to model camera ISP effects; recently extended to 3DGS (Xiao et al., 2025). However, all of these methods significantly increase training time. In contrast, we process the input images using a generalizable multi-view transformer, avoiding scene-specific optimization and preserving 3DGS efficiency.

## 3  METHODOLOGY

We propose a transformer model that takes as input a multi-view sequence of frames with varying appearances (e.g., exposure, white balance, color shifts) and predicts 3D bilateral grids to align each view with the reference frame. In this section, we first review bilateral grid processing (Sec. 3.1), then introduce our transformer architecture (Sec. 3.2) and reference frame selection mechanism (Sec. 3.3). We present our training strategy, leveraging self-supervision from a large feed-forward model, and detail our dataset construction process (Sec. 3.4). Our full pipeline overview is shown in Fig. 2.

### 3.1  PRELIMINARIES

**3D Bilateral Grids for Image Processing.** A 3D bilateral grid (Chen et al., 2007) is a compact data structure suitable for efficient modeling of spatially-varying edge-aware image transformations. It lifts image data into a lower resolution three-dimensional space defined by two spatial coordinates and a guidance dimension derived from the image intensity. By decoupling computational cost from image resolution and preserving semantic edges, bilateral grids enable real-time, and structure-aware processing, making them widely used for tone mapping, stylization, and artifact removal.

In the multi-view setting, we can model the appearance variations using per-view bilateral grids. We denote the $i$-th bilateral grid corresponding to the $i$-th image $\mathbf{I}_i \in \mathbb{R}^{H \times W \times 3}$ as a tensor of local affine transformations $\mathbf{B}_i \in \mathbb{R}^{H_s \times W_s \times D \times 12}$, where $H_s$, $W_s$, and $D$ denote the spatial and guidance dimensions, respectively, such that $(H_s, W_s) << (H, W)$. The last dimension of size 12 corresponds to the flattened parameters of a $3 \times 4$ affine transformation: a $3 \times 3$ matrix $\mathbf{A} \in \mathbb{R}^{3 \times 3}$ and a bias

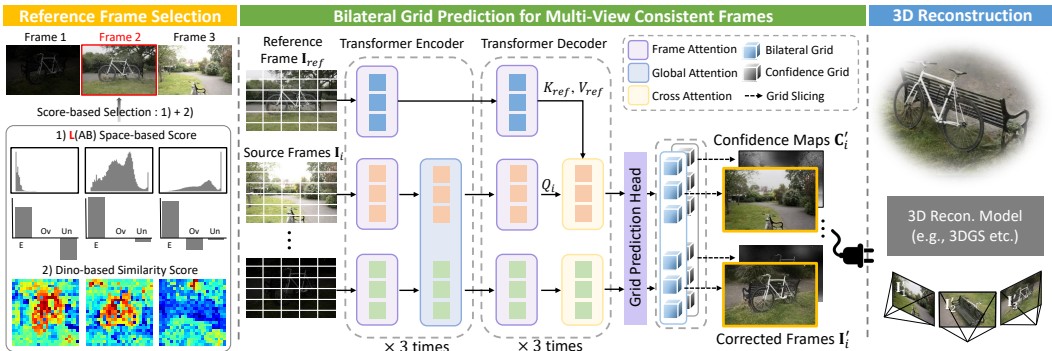

Figure 2: **Architecture Overview**. Our model first patchifies the reference frame $\mathbf{I}_{ref}$ and $N$ input multi-view source images $\{\mathbf{I}_i\}_{i=1}^N$ into tokens. These are passed through the transformer encoder blocks comprising alternating frame-wise and global self-attention layers, repeated 3 times. The decoder uses alternating frame-attention and cross-attention with the reference frame. A final grid prediction head predicts the image and confidence bilateral grids ($\mathbf{B}_i$ and $\mathbf{C}_i$), which are subsequently *sliced* to produce the corrected frames $\{\mathbf{I}'_i\}_{i=1}^N$ and confidence maps $\{\mathbf{C}'_i\}_{i=1}^N$. Based on our reference frame selection which chooses the frame with best photometric quality, we use the resulting harmonized images to train a wide range of 3D reconstruction models.

vector $b \in \mathbb{R}^3$. For input image $\mathbf{I}$, each pixel $d$ with color $I_d \in \mathbb{R}^3$ is transformed to its corresponding output pixel color $I'_d \in \mathbb{R}^3$ by applying the affine transformation: $I'_d = \mathbf{A}_d I_d + b_d$, where $\mathbf{A}_d$ and $b_d$ are the affine parameters specific to pixel $d$. The parameters $\theta_d = (\mathbf{A}_d, b_d)$ are obtained via trilinear interpolation over the neighboring vertices of the bilateral grid:

$$\theta_d = \sum_{i,j,k} w_{ijk}(d)\theta_{ijk}, \tag{1}$$

where $\theta_{ijk} \in \mathbb{R}^{12}$ are the affine parameters at vertex $(i, j, k)$, and $w_{ijk}(d)$ are interpolation weights determined by the spatial and guidance coordinates of pixel $d$. This process is known as *slicing*. For the guidance dimension, we use the pixel luminance following (Chen et al., 2007; Wang et al., 2024a). The bilateral grid resolution is much smaller than the input image resolution, reducing computational cost and preventing the bilateral grid from encoding the high-frequency content of the image.

## 3.2 MULTI-VIEW BILATERAL GRID TRANSFORMER

Our aim is to transform multi-view captures of a scene to be globally consistent, enabling robust 3D reconstruction and novel view synthesis under appearance variations. To achieve harmonization, we propose a multi-view aware transformer predicting per-patch bilateral grid parameters. This approach leverages the conceptual similarity between transformer patch-based processing and the structure of 3D bilateral grids where each vertex encodes a local affine color transformation. By predicting compact grid parameters per patch, which are applied to the original high-resolution images efficiently via lightweight slicing, our model learns spatially-varying image corrections that are consistent across views due to cross-frame attention, while balancing performance and computational cost.

**Model Processing and Outputs.** The input to our model is a sequence of $N$ multi-view frames of a scene $\{\mathbf{I}_i\}_{i=1}^N$ exhibiting potential appearance inconsistencies in color, exposure, white balance, etc. Here, a reference frame $\mathbf{I}_{ref}$, defining the target appearance, is selected via the protocol described in Sec. 3.3, while the remaining frames $\mathbf{I}_{i \neq ref}^N$ serve as source images to be harmonized with $\mathbf{I}_{ref}$. First, each input image $\mathbf{I}_i \in \mathbb{R}^{H \times W \times 3}$ is partitioned into non-overlapping patches $\mathbf{P}_i \in \mathbb{R}^{H_P \times W_P \times J}$, where the number of patches $J$ is $\frac{H}{H_P} \times \frac{W}{W_P}$. Each patch $\mathbf{P}_{i,j}$ is then projected into a feature vector by the patch encoder $\Phi_{\text{embed}}$. These feature vectors, combined with positional encoding to retain the spatial information of each patch, form the input token sequence to our model: $\mathbf{X} = \{\Phi_{\text{embed}}(\mathbf{P}_{i,j}) + \text{PE}_{i,j}\}_{i=1,j=1}^{N,J}$. The input tokens are then processed with the main network $f_\theta$, yielding a set of 3D bilateral grids and confidence grids $\{\mathbf{B}_i\}_{i \neq ref}^N \in \mathbb{R}^{H_s \times W_s \times D \times 12}$, and

$\{\mathbf{C}_i\}_{i \neq ref}^N \in \mathbb{R}^{H_s \times W_s \times D \times 1}$:

$$f_\theta(\mathbf{X}) = \{\mathbf{B}_i\}_{i \neq ref}^N, \{\mathbf{C}_i\}_{i \neq ref}^N, \tag{2}$$

where $\{\mathbf{B}_i\}$ are applied to source frames $\{\mathbf{I}_i\}_{i \neq ref}^N$ to get harmonized frames $\{\mathbf{I}_i'\}$ via slicing (Sec. 3.1).

**Alternating Self and Cross Attention Architecture.** Fig. 2 shows our encoder-decoder transformer architecture, which adopts the alternating attention strategy of VGGT (Wang et al., 2025) to reduce memory cost while preserving the ability to model both intra- and cross-view interactions.

In the *encoder*, each block alternates between *local self-attention*, applied independently within each view to model spatial context and viewpoint-specific patterns, and *global self-attention*, which exchanges information across views at corresponding patch positions.

In the *decoder*, we explicitly align source frames $\{\mathbf{I}_i\}_{i \neq ref}^N$ with the reference frame $\mathbf{I}_{ref}$ to harmonize the appearance. Encoder outputs $\{\mathbf{x}_{i,j}'\}_{i=1}^N$ are separated into $\mathbf{x}_{ref,j}'$ and $\{\mathbf{x}_{i,j}'\}_{i \neq ref}^N$, which are reference and source features, respectively. We replace global-attention layers with *cross-attention* between reference and sources tokens, enabling conditioning on the reference. Specifically, the key $K$ and value $V$ are extracted from $\mathbf{x}_{ref,j}'$, while the query $Q$ is extracted from $\{\mathbf{x}_{i,j}'\}_{i \neq ref}^N$ for each decoder cross-attention block. With this framework, the refined features $\{\mathbf{x}_{i,j}''\}_{i \neq ref}^N$ inherit consistent appearance guided by the reference feature $\mathbf{x}_{ref,j}'$, yielding harmonized multi-view features.

**Bilateral Grid and Confidence Prediction Head.** The decoder's output tokens for each source frame are used to predict the set of bilateral grids $\{\mathbf{B}_i\}_{i \neq ref}^N$ to correct their appearance, rather than directly regressing corrected images. Each token $\{\mathbf{x}_{i,j}''\}_{i \neq ref}^N$ predicts the per-intensity affine transformation parameters of the bilateral grids $\{\mathbf{B}_{i,j}\} \in \mathbb{R}^{D \times 12}$, where $D$ is the intensity guidance dimension. Due to the conceptual similarity between the patch-based transformer model and bilateral grid, we can simply predict each grid-vertex parameters from each token using a small MLP. Per-pixel affine transforms obtained by slicing the resulting grids are applied to the source images obtaining the harmonized images $\{\mathbf{I}_i'\}_{i \neq ref}^N$.

In addition to bilateral grids, we make our model uncertainty-aware by predicting aleatoric uncertainty (Kendall & Cipolla, 2016); modeling inherent noise in the data, e.g., photometric variations in the ground truths or information loss in over-/under-exposed regions; thus stabilizing the training loss. Since we cannot directly obtain the confidence maps from our prediction head, as this would require a dense prediction head (Wang et al., 2025), we instead predict a low-resolution confidence grid along with each bilateral grid $\{\mathbf{C}_i\} \in \mathbb{R}^{H_s \times W_s \times D \times 1}$. Thus, for each patch position $j$, the grid prediction head outputs $\{\mathbf{C}_{i,j}\} \in \mathbb{R}^{D \times 1}$. Applying the slicing operation as before using the source images, we obtain full-resolution confidence maps $\{\mathbf{C}_{i,j}'\} \in \mathbb{R}^{H \times W \times 1}$ that reflect the confidence of each pixel.

## 3.3 REFERENCE FRAME SELECTION

We employ a reference-frame-based strategy for two practical reasons: it maintains a consistent color space across frames, mitigating drift, and it allows explicit control over the final reconstruction appearance by specifying the reference frame. However, naively using the first or a random frame as the reference poses the risk of selecting an outlier, potentially causing drift or degrading quality, as shown in previous studies (Ren et al., 2020; Lee et al., 2022; Shin et al., 2025). To address this, we propose a method to select a reference frame that is both photometrically reliable and semantically representative at inference, as visualized in Fig. 2 (left).

We assess the semantic representativeness of each frame by computing cosine similarities of DINOv2 embeddings (Oquab et al., 2023). This yields a per-frame similarity score $S_{\text{DINO}}$, favoring those that contain rich scene information. However, as DINOv2 is robust to illumination variations, underexposed frames may also receive undesirably high similarity scores. Thus, we also assess the photometric quality of frames with respect to under-/over-exposure using the luminance channel $L$ in CIE-LAB color space. We penalize frames with extreme over-/under-exposure ratios (e.g., below 5% or above 95%) and combine this with the normalized entropy of the luminance distribution to form:

$$S_{\text{LAB}} = \lambda_{ent}\left(-\sum_l p(l) \log p(l)\right) + \lambda_{ov}\frac{1}{|L|}\sum_{i,j}[L_{ij} \geq 250] + \lambda_{un}\frac{1}{|L|}\sum_{i,j}[L_{ij} \leq 5], \tag{3}$$

where, $p(l)$ is the probability of intensity level $l$ estimated from the normalized luminance histogram, and $\lambda_{ent}$, $\lambda_{ov}$, and $\lambda_{un}$ are 1, $-0.5$, and $-0.5$, respectively. We define the overall score for frame $i$ as $S_i = \alpha \cdot S_{\text{LAB},i} + (1 - \alpha) \cdot S_{\text{DINO},i}$ and the frame with the highest $S_i$ is chosen as the reference, where $\alpha$ is set to 0.5. This pipeline maintains interpretability of reference-based harmonization while enabling automatic and robust selection.

## 3.4 TRAINING STRATEGY

**Training Objectives with Self-Supervised Guidance.** We train our model with the following probabilistic loss function (Kendall & Cipolla, 2016) to predict the corrected images:

$$\mathcal{L}_{conf} = \sum_{i=1}^{N} \mathbf{C}'_i \odot \|\hat{\mathbf{I}}_i - \mathbf{I}'_i\|_1 - \beta \log(\mathbf{C}'_i), \tag{4}$$

where $\mathbf{I}'_i$ is the source image corrected with the bilateral grid $\mathbf{B}_i$. This probabilistic loss modulates the L1 loss between the corrected image $\mathbf{I}'_i$ and the ground-truth image $\hat{\mathbf{I}}_i$, allowing the model to rely less on its predictions in challenging areas where image detail recovery is difficult. Along with $\mathcal{L}_{conf}$, we apply the total variation loss, $\mathcal{L}_{TV}$, encouraging smoothness of the bilateral grids:

$$\mathcal{L}_{TV} = \frac{1}{N} \sum_{i=1}^{N} \sum_{h,w,d} \left( \|\Delta_h \mathbf{B}_i(h,w,d)\|_2^2 + \|\Delta_w \mathbf{B}_i(h,w,d)\|_2^2 + \|\Delta_d \mathbf{B}_i(h,w,d)\|_2^2 \right), \tag{5}$$

where $\Delta$ denote forward difference operators.

As it is not straightforward to obtain real-world paired data to train our model, we further introduce a self-supervised loss. This enables training in cases without ground truth images and helps enforce view-consistency across frames. We leverage a large, pretrained feed-forward 3D reconstruction model $h_\theta$ (e.g., AnySplat (Jiang et al., 2025)). Specifically, given a sequence of frames on each training iteration, we set the first frame as reference ($ref = 0$) and predict camera poses $\{p_i\}_{i=0}^{N}$ and Gaussian primitives $\{\mathbf{G}_i\}_{i=0}^{N}$ from all the frames, including the first input frame $\mathbf{I}_{ref}$ and the remaining corrected frames $\{\mathbf{I}'_i\}_{i=1}^{N}$.

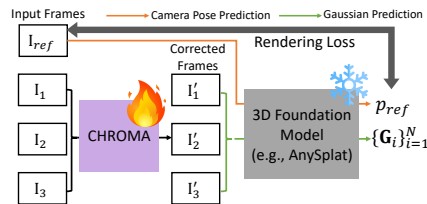

Figure 3: 3D foundation model-based self-supervised loss pipeline.

The predicted Gaussians for non-reference frames $\{\mathbf{I}'_i\}_{i=1}^{N}$ are then reprojected to the viewpoint of the reference frame using predicted camera pose $p_{ref}$, and we compute the perceptual loss utilizing VGG features (Simonyan & Zisserman, 2014) between the original reference frame $\mathbf{I}_{ref}$ and the rendered image. This encourages the outputs of our bilateral transformer network $f_\theta$ to remain in a consistent color space, so that the downstream 3D reconstruction model $h_\theta$ can build a coherent scene representation and generate stable novel views. Based on this, the self-supervised consistency loss $\mathcal{L}_{ss}$ is defined as:

$$\mathcal{L}_{ss} = \text{VGG}(\mathbf{I}_{ref} - Rasterizer(p_{ref}, \{\mathbf{G}_i\}_{i=1}^{N})), \quad \text{where} \quad \{p_i, \{\mathbf{G}_i\}\}_{i=0}^{N} = h_\theta\left(f_\theta(\{\mathbf{I}_i\}_{i=0}^{N})\right), \tag{6}$$

with *Rasterizer* denoting the differentiable Gaussian rasterizer that projects the set of predicted Gaussian primitives $\{\mathbf{G}_i\}$ into the viewpoint $p_{ref}$ of the reference frame. Thus, our total loss is:

$$\mathcal{L} = \mathcal{L}_{conf} + \lambda_{tv}\mathcal{L}_{TV} + \lambda_{ss}\mathcal{L}_{ss}, \tag{7}$$

where $\lambda_{tv}$ and $\lambda_{ss}$ are set to 0.5 and 0.1, respectively. Note that we use the predicted Gaussian opacity to mask out unreliable regions in both the reference image $\mathbf{I}_{ref}$ and the rasterized image from the predicted Gaussians $\{\mathbf{G}_i\}$, allowing us to apply $\mathcal{L}_{ss}$ only to well-reconstructed areas.

**Training Datasets.** Acquiring large-scale multi-view paired data with realistic appearance variations is extremely challenging. Our approach handles this limitation by (i) simulating diverse and controllable variations in a synthetic setting, and (ii) introducing a self-supervised rendering loss that allows us to effectively learn from unpaired real-world data (Eq. 6). For paired data, we use multi-view consistent sequences from the DL3DV dataset (Ling et al., 2024), comprising 10K scenes, each containing more than 300 frames. For each scene we synthesize realistic appearance variations

simulating camera ISP pipelines, thereby creating input-target pairs with controllable appearance changes. For unpaired data, we use the WildRGB-D dataset (Xia et al., 2024), which captures objects under 360 degree rotations. It consists of 8K recorded objects, where each scene also contains roughly 300 frames. As the viewpoint changes, the environmental lighting and camera's auto-ISP settings induce appearance variations, such as fluctuating exposure, white balance and illumination, providing a realistic source of real-world inconsistencies. During training, the supervised loss $\mathcal{L}_{conf}$ is applied only to paired data (DL3DV), while our self-supervised consistency loss $\mathcal{L}_{ss}$ and total variation loss $\mathcal{L}_{tv}$ are jointly optimized over both datasets. This enables our model to generalize well across both synthetic and real variations.

**Parametric Camera ISP Simulation.** We apply parametric camera ISP simulation on top of the DL3DV dataset (Ling et al., 2024). Building on the unprocessing framework of (Brooks et al., 2019), we reverse the camera pipeline to obtain linear RGB images and apply randomized variations in white balance, exposure, digital gain, gamma perturbation, and color correction matrices (CCMs). To realistically simulate illumination diversity, we model the exposure distribution as a bi-modal mixture representing both daytime and nighttime conditions. Specifically, exposure values $e$ are drawn from:

$$e \sim \pi \mathcal{N}(\mu_{\text{day}}, \sigma_{\text{day}}^2) + (1 - \pi)\mathcal{N}(\mu_{\text{night}}, \sigma_{\text{night}}^2),$$

where $(\mu_{\text{day}}, \sigma_{\text{day}})$ and $(\mu_{\text{night}}, \sigma_{\text{night}})$ are the mean and variance of day and nighttime exposures respectively, and $\pi \in [0, 1]$ controls the prior probability of day and night; set to $(1.0, 0.2)$, $(0.2, 0.1)$, $0.5$, respectively. This simulates both well-lit and low-light scenes, enhancing robustness to ISP-induced inconsistencies under varying illumination conditions, as illustrated in Fig. 1(a).

**Implementation Details.** As Fig. 2 shows, our transformer encoder employs 3 layers of alternating frame-wise and global-attention. The decoder comprises 3 layers of alternating frame-wise and cross-attention. The model is relatively compact, with 137.84M parameters in total. We train by optimizing Eq. 7 with AdamW (Loshchilov & Hutter, 2017) for 70K iterations. We use dynamic batch loading to randomly sample between 4 and 24 frames from a selected multi-view scene (Sec. 3.4). Input images are resized to $224 \times 224$ resolution with patch size $8 \times 8$, resulting in a total of $28 \times 28 \times 8$ bilateral grid vertices per frame; one for each input image patch with a guidance dimension of $D = 8$. Training takes roughly a day with current GPU hardware (60 TFLOPS fp32 and 80 GB of memory).

## 4 EXPERIMENTS

We evaluate our method under three types of appearance variations: (a) *ISP variations*, (b) *exposure changes*, and (c) *real-world capture conditions*. This section describes the datasets used, baseline comparison methods, evaluation metrics, followed by detailed experimental results and ablations.

**Evaluation Datasets.** For (a) *ISP variations*, we use a camera ISP variation dataset derived from the DL3DV dataset (Ling et al., 2024). As described in Sec. 3.4, this includes white balance, exposure, gamma, and CCM adjustments. We evaluate on a diverse set of 25 held-out scenes which vary in content (indoor/outdoor), spatial characteristics (bounded/unbounded), and lighting conditions, as shown in the Supplementary. For (b) *exposure variations*, we use the MipNeRF360-VE dataset released by Luminance-GS (Cui et al., 2025), which is based on the unbounded MipNeRF 360 dataset (Barron et al., 2022) with varying exposure and gamma correction. For (c) *real-captured scenes*, we evaluate on the real-world captured BilaRF dataset (Wang et al., 2024a), comprising mainly of nighttime scenes captured with flash illumination, posing real-world appearance shifts.

**Baselines.** We compare against state-of-the-art methods that incorporate appearance modeling into 3DGS: WildGaussians (Kulhanek et al., 2024), GS-W (Zhang et al., 2024), Luminance-GS (Cui et al., 2025), and 3DGS-4DBAG (Xiao et al., 2025) which jointly optimizes 3DGS and 4D bilateral grids. We also compare to vanilla 3DGS (Kerbl et al., 2023) (fast version from Taming-GS (Mallick et al., 2024)), 2DGS (Huang et al., 2024a), and DashGS (Chen et al., 2025).

**Metrics.** Quantitative evaluation is conducted using PSNR, SSIM, and LPIPS. When using appearance embeddings, the reconstructed color space may differ from the ground truth, leading to unfairly low scores despite accurate geometry. To address this, per-channel global affine color correction (CC) is applied to the rendered images, following (Wang et al., 2024a; Xiao et al., 2025; Mildenhall et al., 2021). Color-corrected metrics provide a more reliable measure of geometric quality under color discrepancies. Processing time on a mid-range GPU is also reported, including bilateral grid

Table 1: Comparison of our model combined with baseline 3D reconstruction methods, the baselines alone, and approaches using appearance embeddings. **CC** indicates per-channel affine color correction. Reported times include scene fitting; for our model, they also include processing overhead.

| *Dataset* Method | PSNR ↑ | **PSNR CC** ↑ | SSIM ↑ | **SSIM CC** ↑ | LPIPS ↓ | **LPIPS CC** ↓ | Time (s) ↓ |
|---|---|---|---|---|---|---|---|
| *DL3DV dataset w/ ISP variation* | | | | | | | |
| 2DGS (Huang et al., 2024a) | 22.78 | 26.81 | 0.8496 | 0.8608 | 0.2829 | 0.2169 | 13m 48s |
| 3DGS (Kerbl et al., 2023) | 21.43 | 26.25 | 0.8553 | 0.8749 | 0.2712 | 0.2069 | 3m 39s |
| DashGS (Chen et al., 2025) | 23.35 | 28.17 | 0.8916 | 0.9029 | 0.2357 | 0.1782 | 3m 12s |
| WildGaussians (Kulhanek et al., 2024) | 18.15 | 24.08 | 0.7663 | 0.8188 | 0.3050 | 0.2567 | 2h 10m 31s |
| GS-W (Zhang et al., 2024) | 19.34 | 26.29 | 0.7910 | 0.8420 | 0.3092 | 0.2375 | 35m 54s |
| Luminance-GS (Cui et al., 2025) | 20.00 | 26.14 | 0.7962 | 0.8466 | 0.2975 | 0.2290 | 14m 29s |
| **2DGS + Ours** | 24.97 | 26.92 | 0.8564 | 0.8621 | 0.2312 | 0.2236 | 14m 3s |
| **3DGS + Ours** | 25.26 | 27.28 | 0.8650 | 0.8753 | 0.2191 | 0.2118 | 3m 55s |
| **DashGS + Ours** | 26.45 | 28.92 | 0.8953 | 0.9035 | 0.1794 | 0.1703 | 3m 27s |
| *MipNeRF360-VE* | | | | | | | |
| 2DGS (Huang et al., 2024a) | 16.75 | 22.03 | 0.5588 | 0.6724 | 0.3646 | 0.3622 | 25m 38s |
| 3DGS (Kerbl et al., 2023) | 16.50 | 21.00 | 0.5896 | 0.6715 | 0.3432 | 0.3517 | 4m 7s |
| DashGS (Chen et al., 2025) | 17.01 | 21.94 | 0.6043 | 0.7052 | 0.3212 | 0.3221 | 3m 3s |
| WildGaussians (Kulhanek et al., 2024) | 18.90 | 25.35 | 0.6470 | 0.7278 | 0.3261 | 0.3193 | 2h 30m 3s |
| GS-W (Zhang et al., 2024) | 15.66 | 25.81 | 0.5256 | 0.7580 | 0.3385 | 0.2912 | 48m 45s |
| Luminance-GS (Cui et al., 2025) | 18.12 | 23.12 | 0.6641 | 0.7352 | 0.3043 | 0.2851 | 22m 16s |
| **2DGS + Ours** | 18.99 | 26.37 | 0.7528 | 0.8125 | 0.2610 | 0.2446 | 25m 50s |
| **3DGS + Ours** | 18.34 | 26.25 | 0.7554 | 0.8149 | 0.2592 | 0.2428 | 4m 19s |
| **DashGS + Ours** | 18.19 | 26.21 | 0.7507 | 0.8131 | 0.2610 | 0.2435 | 3m 14s |
| *BilaRF dataset* | | | | | | | |
| 2DGS (Huang et al., 2024a) | - | 23.43 | - | 0.746 | - | 0.308 | 13m 26s |
| 3DGS (Kerbl et al., 2023) | - | 24.23 | - | 0.8019 | - | 0.2594 | 3m 43s |
| DashGS (Chen et al., 2025) | - | 24.34 | - | 0.7880 | - | 0.2607 | 3m 46s |
| WildGaussians (Kulhanek et al., 2024) | - | 23.19 | - | 0.7424 | - | 0.3121 | 1h 58m 06s |
| GS-W (Zhang et al., 2024) | - | 24.94 | - | 0.8056 | - | 0.2764 | 40m 34s |
| Luminance-GS (Cui et al., 2025) | - | 23.41 | - | 0.7931 | - | 0.2750 | 18m 40s |
| 2DGS-4DBAG (Xiao et al., 2025) | - | 24.80 | - | 0.773 | - | 0.273 | - |
| 3DGS-4DBAG (Xiao et al., 2025) | - | 24.90 | - | 0.774 | - | 0.256 | - |
| **2DGS + Ours** | - | 25.27 | - | 0.8147 | - | 0.2499 | 13m 30s |
| **3DGS + Ours** | - | 25.60 | - | 0.8240 | - | 0.2368 | 3m 48s |
| **DashGS + Ours** | - | 26.25 | - | 0.8356 | - | 0.2158 | 3m 50s |

Table 2: Comparison with 2D image correction methods on MipNeRF360-VE (Cui et al., 2025). Motion smoothness and temporal flickering from VBench (Huang et al., 2024b) are reported as the most relevant metrics for video quality under photometric correction.

| Comparison with 2D Works | PSNR CC ↑ | SSIM CC ↑ | LPIPS CC ↓ | motion smoothness ↑ | temporal flickering ↑ | time ↓ |
|---|---|---|---|---|---|---|
| CoTF (Li et al., 2024) | 23.15 | 0.7573 | 0.2698 | 0.8591 | 0.8364 | 3.46 |
| MSEC (Afifi et al., 2021) | 23.98 | 0.7169 | 0.3359 | 0.8140 | 0.7848 | 7.79 |
| MSLTNet (Zhou et al., 2024) | 24.56 | 0.7697 | 0.2697 | 0.8248 | 0.7974 | 3.12 |
| UEC (Cui et al., 2024) | 25.98 | 0.8143 | **0.2416** | 0.8415 | 0.8124 | 7.64 |
| **Ours** | **26.25** | **0.8149** | 0.2428 | **0.8707** | **0.8428** | 8.59 |

inference and reference frame selection for our method. We note that all reported inference times include the total time for frame processing and 3DGS reconstruction, and are measured on the same GPU architecture to ensure fairness and consistency across methods.

## 4.1 RESULTS

Table 1 reports quantitative results across all three datasets, with qualitative examples in Fig. 4. For all scenes, we employ our reference frame selection strategy (Sec. 3.3); no ground-truth image was used as reference for fair comparison. Despite the distinct characteristics of the three datasets, our method combined with 2DGS, 3DGS, or DashGS significantly outperforms each baseline, with fewer artifacts, and matches or surpasses per-scene appearance optimization methods. This demonstrates the robustness and generalizability of our model, and highlights the multi-view consistency leading to improved reconstructions. Moreover, methods with joint optimization of geometry and appearance introduce significant latency; more than doubling the overall training time. Notably, our model efficiently processes large-scale inputs, handling over 300 frames in a single forward pass. In practice,

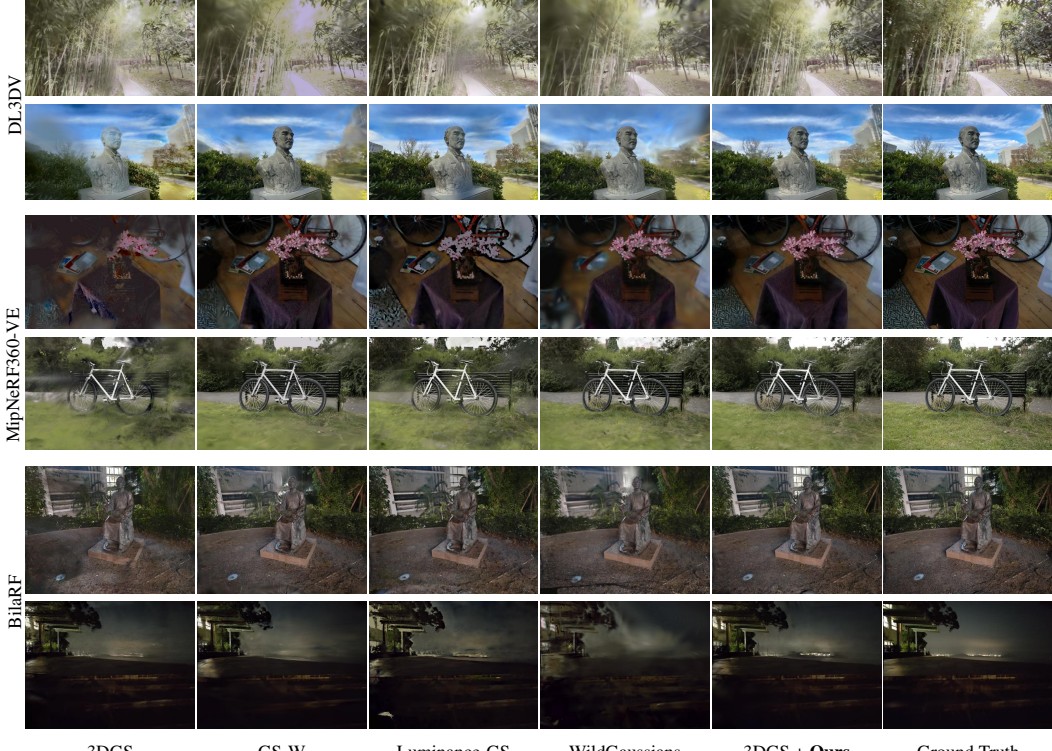

|  |  |  |  |  |  |
| :---: | :---: | :---: | :---: | :---: | :---: |
| 3DGS | GS-W | Luminance-GS | WildGaussians | 3DGS + **Ours** | Ground Truth |

Figure 4: Qualitative results grouped by dataset: DL3DV, MipNeRF360-VE, and BilaRF.

inference takes only about 2-3 seconds for BilaRF (roughly 30-70 images per scene), and up to 15 seconds for DL3DV and MipNeRF360-VE, each containing more than 200 frames per scene.

Table 2 compares our method with 2D image exposure correction baselines prior to running 3DGS. Since these methods primarily target exposure adjustment, we conduct the comparison on MipN-eRF360-VE dataset, containing mostly exposure-related variations. While these methods perform well on single frames, they lack multi-view consistency, resulting in degraded 3DGS performance. Similar to our method, UEC (Cui et al., 2024) uses an input reference exposure frame, but still operates independently per frame, leading to lower overall performance. For fair comparison, we use the same reference frames for both our method and UEC, selected by our reference frame pipeline (Sec. 3.3). Since MipNeRF360-VE scenes contain over 200 frames per scene and our model uses a transformer architecture that processes all views jointly, CHROMA incurs a few additional seconds of overhead compared to CNN-based 2D methods that operate per frame. However, this overhead is small relative to the overall 3DGS reconstruction time, which takes several minutes, so the additional cost does not meaningfully affect the total pipeline runtime.

Table 3: Ablation study. (Top) MipNeRF360-VE dataset (Bottom) BilaRF dataset.

| Ablation | PSNR CC | SSIM CC | LPIPS CC |
| :---: | :---: | :---: | :---: |
| single-frame processing | 26.19 | 0.8131 | 0.2435 |
| random reference frame | 25.11 | 0.7559 | 0.3089 |
| DINO-based reference frame | 25.95 | 0.7871 | 0.2758 |
| Ours (full) | **26.25** | **0.8149** | **0.2428** |
| w/o self-supervised loss | 25.21 | **0.8245** | 0.2375 |
| L1-based self-supervised loss | 25.28 | 0.8227 | 0.2377 |
| Ours (w/ self-supervised loss) | **25.60** | 0.8240 | **0.2368** |

We conduct ablations to isolate the contribution of each model component. Table 3 reports results on single vs multi-frame processing, reference frame selection, and the self-supervised loss. For experiments on multi-frame processing and reference frame selection, we use MipNeRF360-VE dataset, which exhibits large exposure variations across frames.

**Single-frame processing:** We observe that processing single-frame inputs independently yields slightly lower performance than processing the entire sequence in a single step, highlighting the importance of our transformers ability to leverage cross-view information. **Reference frame selection:** Simply choosing the first frame as the reference, which may be badly exposed or contain little

semantic information (e.g. sky), significantly degrades performance. While DINO features help capture semantic similarity, the results show that relying on them alone is insufficient to robustly handle appearance variations, underscoring the importance of our dedicated reference frame selection mechanism. **Self-supervised loss:** To assess the effect of the self-supervised consistency loss $\mathcal{L}_{ss}$, we evaluate on the held-out BilaRF dataset, which contains unpaired scenes with real-world variations. Incorporating $\mathcal{L}_{ss}$ enables us to train on real-world unpaired data from the WildRGB-D dataset. This helps bridge the domain gap from synthetic ISP variations (paired DL3DV data) to real-world appearance variations and yields clear gains in real-data performance. In addition, our ablation shows that the VGG-based perceptual loss serves as a more stable and informative regularizer than a pixel-wise L1 loss. The VGG variant is more robust to low-level photometric noise and misalignment, and it prevents the model from overfitting to minor detail inconsistencies in AnySplat's renderings, leading to noticeably stronger performance.

## 5 CONCLUSIONS

We have presented **CHROMA**, a feed-forward transformer framework for multi-view consistent harmonization via bilateral grid prediction. By explicitly enforcing photometric consistency across views, CHROMA integrates with existing 3D reconstruction pipelines without scene-specific optimization. Leveraging a large pretrained feed-forward model, we enable learning from unpaired data with self-supervision, improving robustness to real-world variations. Our reference frame selection strategy identifies a representative frame that is both photometrically and semantically reliable. Our experiments show that CHROMA consistently outperforms per-scene appearance embedding baselines, achieving higher reconstruction quality while reducing training overhead.

**Future Work.** Our model generalizes well to real-world appearance variations, and we plan to extend it to fully in-the-wild scenarios by modeling transient objects and integrating with larger feed-forward reconstruction frameworks (Wang et al., 2025; Jiang et al., 2025; Keetha et al., 2025). Unlike recent scene-specific in-the-wild methods (e.g., Kulhanek et al. (2024); Zhang et al. (2024); Wang et al. (2024b); Bai et al. (2025)), embedding our harmonization module into these models could enable a generalizable, feed-forward approach for 3D reconstruction from unconstrained photo collections.

**Limitations.** Unlike scene-specific approaches that optimize 3DGS and jointly learn per-view appearance embeddings, such as Luminance-GS (Cui et al., 2025) or GS-W (Zhang et al., 2024), our method requires training a dedicated network for image harmonization, prior to downstream reconstruction. However, once trained, our method provides scene-agnostic, feed-forward harmonization and can process hundreds of images in a single pass without any scene-specific optimization. Additionally, our method focuses on correcting camera ISP-induced photometric inconsistencies and does not explicitly handle effects such as specular highlights or reflections. Because bilateral grid transformations are generally smooth, they cannot model high-frequency changes caused by reflectance or strong specularities, which remains a limitation of our approach.

ACKNOWLEDGMENTS

This work was partially supported by the National Research Foundation of Korea(NRF) grant funded by the Korea government(MSIT) (RS-2024-00338439), Institute for Information & Communications Technology Planning & Evaluation (IITP) grant funded by the Korea government(MSIT) (No.RS-2025-25441838 and RS-2024-00457882).

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
