

Figure 5: Training dataset generation examples using DL3DV scenes.

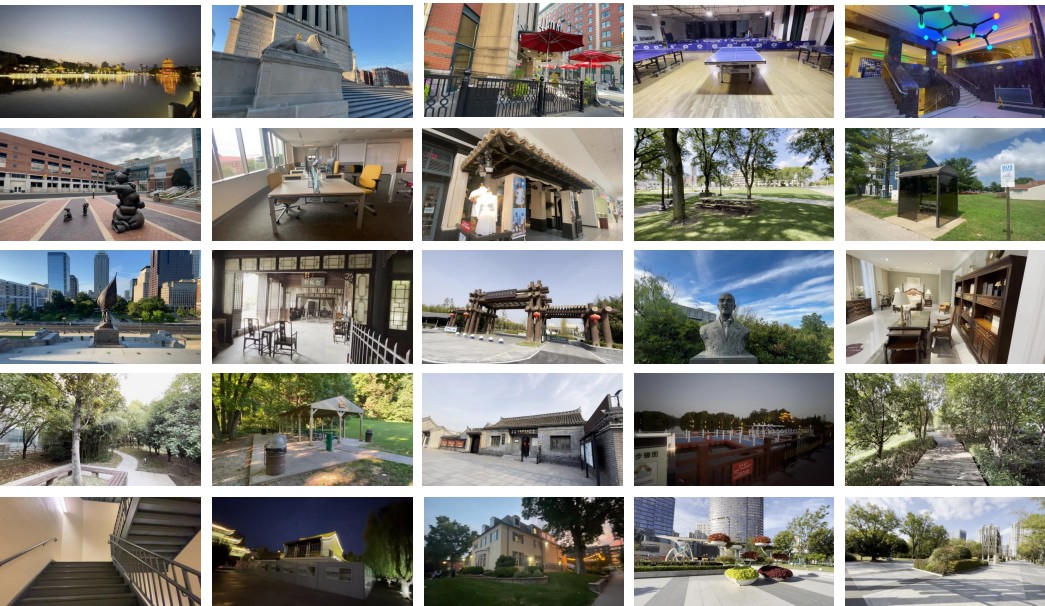

Figure 6: Examples from the DL3DV inference set.

## A  SUPPLEMENTARY

This supplementary material provides further detailed explanations of our model structure and implementation details. We also provide additional quantitative and qualitative comparisons to support effectiveness of our method.

### A.1  DATASET GENERATION

In this section, we specify the details of the hyperparameters used to build our training dataset from DL3DV (Ling et al., 2024) real-world scenes, following the reverse ISP process of Brooks *et al.* (Brooks et al., 2019).

For parametric augmentations using the reverse ISP process (Fig. 5, right), we use a modified implementation (https://github.com/aasharma90/UnprocessDenoising_PyTorch), and we randomly sample hyperparameters controlling digital gain, exposure, gamma value, and shadow and highlight thresholds. Along with the bi-modal mixture distribution of exposure explained in Sec 3.4, digital gain and gamma values are uniformly sampled from the following ranges:

Table 4: **Detailed network structure.** The input to the transformer is a multi-view image sequence, where the first frame is the reference image. B denotes the batch size and S is the number of input frames. The transformer output is a set of 3D bilateral grids, which are subsequently sliced and applied to the original source images, excluding the reference frame. For the output of 1-1, 768 represents the channel dimension, and 784 corresponds to the number of patches.

| IDX | STRUCTURE | INPUT | OUTPUT |
|---|---|---|---|
| **Patch Embedding** | | | |
| 1-1 | CONCAT(DINOv1, Conv) | Input(B $\times$ S $\times$ 3 $\times$ H(224) $\times$ W(224)) | B $\times$ S $\times$ 768 $\times$ 784 |
| **Transformer Encoder** | | | |
| 2-1 | FrameAttention | OUT1-1 | B $\times$ S $\times$ 768 $\times$ 256 |
| 2-2 | GlobalAttention | OUT2-1 | B $\times$ S $\times$ 768 $\times$ 256 |
| 2-3 | FrameAttention | OUT2-2 | B $\times$ S $\times$ 768 $\times$ 256 |
| 2-4 | GlobalAttention | OUT2-3 | B $\times$ S $\times$ 768 $\times$ 256 |
| 2-5 | FrameAttention | OUT2-4 | B $\times$ S $\times$ 768 $\times$ 256 |
| 2-6 | GlobalAttention | OUT2-5 | B $\times$ S $\times$ 768 $\times$ 256 |
| **Transformer Decoder** | | | |
| 3-1 | CrossAttention | OUT2-6 | B $\times$ S-1 $\times$ 768 $\times$ 256 |
| 3-2 | FrameAttention | OUT3-1 | B $\times$ S-1 $\times$ 768 $\times$ 256 |
| 3-3 | CrossAttention | OUT3-2 | B $\times$ S-1 $\times$ 768 $\times$ 256 |
| 3-4 | FrameAttention | OUT3-3 | B $\times$ S-1 $\times$ 768 $\times$ 256 |
| 3-5 | CrossAttention | OUT3-4 | B $\times$ S-1 $\times$ 768 $\times$ 256 |
| 3-6 | FrameAttention | OUT3-5 | B $\times$ S-1 $\times$ 768 $\times$ 256 |
| **Grid Prediction Head** | | | |
| 4-1 | (MLP, Tanh) | OUT3-6 | B $\times$ S-1 $\times$ 512 $\times$ 256 |
| 4-2 | (MLP, Tanh) | OUT4-1 | B $\times$ S-1 $\times$ 256 $\times$ 256 |
| 4-3 | MLP | OUT4-2 | B $\times$ S-1 $\times$ 104 $\times$ 256 |
| 4-4-1 | (Tanh, Reshape) | OUT4-3[:, :, :-8, :] | (B * S-1) $\times$ 12 $\times$ 8 $\times$ 28 $\times$ 28 |
| 4-4-2 | (Tanh, Reshape) | OUT4-3[:, :, -8:, :] | (B * S-1) $\times$ 1 $\times$ 8 $\times$ 28 $\times$ 28 |
| **Slicing & Applying** | | | |
| 5-1 | Slicing & Applying | (OUT4-4-1, Input) | (B * S-1) $\times$ 3 $\times$ H $\times$ W |
| 5-2 | Slicing | OUT4-4-2 | (B * S-1) $\times$ 1 $\times$ H $\times$ W |

1. Red gain, blue gain: $\mathcal{N}(\mu_{\text{gain}}, \sigma_{\text{gain}}^2)$

2. Gamma: $[0.5, 3.0]$,

where $\mu_{\text{gain}}, \sigma_{\text{gain}}$ are set to $1.0$ and $0.2$, respectively.

Additionally, we apply random $3 \times 3$ color correction matrices (CCMs) to the images. In order to generalize to multiple cameras, we sample random convex combinations from (Brooks et al., 2019), which includes four CCMs: A Sony A7R (full-frame), an Olympus OMD E-M10 (Micro Four-Thirds), a Sony RX100 IV (1 inch) and a Nexus 6P (1/2.3 inch).

We visualize some examples from our training dataset that are generated based on DL3DV, a large-scale scene dataset (Ling et al., 2024), in Fig. 5.

Fig. 6 visualizes scenes from our DL3DV inference set used in the main experiments. We carefully curate this set to include a wide range of categories, including: nature & outdoors, tourist attractions, restaurants and cafes, sports & fitness, parks & recreation, educational institutions, events & conferences, tech & business, shopping centers, and transportation hubs. This inference set allows us to assess the models performance across diverse real-world environments and application domains.

Table 5: **Runtime and peak GPU memory usage across different numbers of input frames.** Runtime is measured in seconds and GPU memory usage is reported in gigabytes.

| Input Frames | 2 | 4 | 8 | 10 | 20 | 50 | 100 | 200 |
|---|---|---|---|---|---|---|---|---|
| Time (s) | 0.0786 | 0.1028 | 0.1451 | 0.1779 | 0.2948 | 0.8172 | 2.1219 | 6.4382 |
| Memory (GB) | 2.1384 | 2.1608 | 2.3142 | 2.4020 | 2.8225 | 4.1070 | 6.2593 | 10.5508 |

Table 6: **Inference time comparison between predicting images and bilateral grids.** Runtime is measured in seconds. OOM (Out Of Memory) indicates that the system ran out of memory and terminated the process.

| Input Resolution | 256 | 384 | 512 | 768 | 1K | 2K | 4K |
|---|---|---|---|---|---|---|---|
| Dense Prediction Head | 0.0897 | 0.1390 | 0.2451 | 0.8761 | OOM | OOM | OOM |
| Bilateral Grid Head | 0.0791 | 0.0818 | 0.0804 | 0.0862 | 0.0880 | 0.1347 | 0.3783 |

## A.2 Network Details

We provide the detailed structure of our network in Tab. 4, relating to the transformer architecture discussed in Sec.3.2 of the main paper. Also, as shown in Tab. 5, we evaluate the inference runtime and peak GPU memory usage for different numbers of input frames. These measurements are conducted on a single NVIDIA RTX 3090 GPU using FlashAttention v2 (Dao, 2023), with the image resolution fixed at $224 \times 224$.

## A.3 Direct Image Prediction vs Bilateral Grid Estimation

In this section, we demonstrate the advantages of our proposed method that predicts 3D bilateral grids from the patches of the transformer model, rather than predicting the corrected images directly. We conduct an additional experiment making a modification to our model, in which the output images are directly decoded from the patches by the transformer. Note that while predicting the bilateral grid from each patch can be achieved with relatively shallow layers, directly regressing the full-resolution image requires a more complex architecture.

To demonstrate this, we adopt the dense prediction head (DPT) of VGGT (Wang et al., 2025), which is used to predict depth maps, by modifying it for the image prediction setting. To ensure a fair comparison, we adjust the number of layers and channels in the head to make the total number of parameters comparable with ours ($\sim 140$M). This approach, however, introduces significant memory overhead and constrains the output resolution, making it infeasible to predict at high resolutions on a mid-grade GPU, as shown in Tab. 6 and Tab. 7. Predicting images beyond 1K resolution causes out-of-memory issues on a NVIDIA RTX 3090 GPU. Tab. 6 compares the inference time (in seconds) using the two different prediction heads across a range of input image resolutions from 224 to 4K. Inference time using the DPT is slower than the bilateral grid for all input resolutions and increases sharply with resolution. At low resolutions (e.g., 224, 384), both heads are comparable, but the bilateral grid head is slightly faster. Tab. 7 compares the peak GPU memory usage (in GB) for two architectures, showing how each method scales in memory consumption. The bilateral grid head scales better thanks to low-resolution processing and interpolation.

Additionally, we conduct a quantitative performance comparison between the DPT-based model and our bilateral grid-based model. We modify the DPT model to predict the residual between

Table 7: **Peak memory usage comparison between predicting images and bilateral grids.** GPU memory usage is reported in gigabytes. OOM (Out Of Memory) indicates that the system ran out of memory and terminated the process.

| Input Resolution | 256 | 384 | 512 | 768 | 1K | 2K | 4K |
|---|---|---|---|---|---|---|---|
| Dense Prediction Head | 2.2034 | 2.6398 | 3.9537 | 11.6560 | OOM | OOM | OOM |
| Bilateral Grid Head | 2.1425 | 2.1461 | 2.1519 | 2.1730 | 2.1996 | 2.6300 | 4.5519 |

Table 8: Performance comparison between direct regression of image corrections using a DPT head and our bilateral grid transformer on the BilaRF dataset.

| Method | PSNR CC | SSIM CC | LPIPS CC |
|---|---|---|---|
| Dense Prediction Head | 24.01 | 0.7291 | 0.3882 |
| Bilateral Grid Head | 25.60 | 0.8240 | 0.2368 |

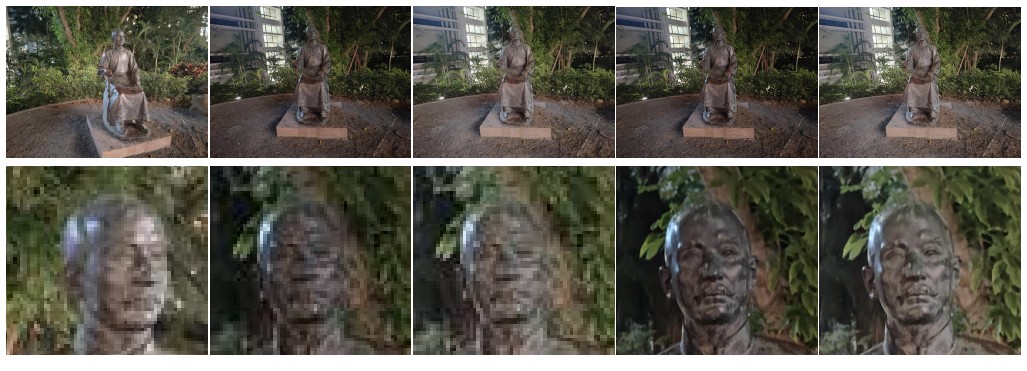

| Reference Image | Source (375 × 500) | Corrected (375 × 500) | Source (3K × 4K) | Corrected (3K × 4K) |

Figure 7: Visualization of learned bilateral grid application to low- and high-resolution images. Left: reference frame; middle: low-resolution (375 × 500) source and corrected images; right: high-resolution (3K × 4K) counterparts. Best viewed when zoomed in.

the input and ground-truth images rather than directly regressing the corrected images, following prior work (Song et al., 2021). Despite this adjustment, its reconstruction performance remains significantly below that of our bilateral grid head, as shown in Tab. 8. This can be explained as follows. Unlike our bilateral grid framework, dense image-to-image translation is a much more complex task and requires substantial architectural engineering to achieve competitive performance. Full-resolution image regression typically relies on multi-scale skip connections, feature pyramids, or residual designs optimized for reconstruction tasks. Without such specialized components, a dense prediction head struggles to capture high-frequency structures and fine photometric cues, which explains its suboptimal performance in our controlled comparison.

There is an additional disadvantage when training the DPT-based model, namely that the dense prediction head is inherently resolution-fixed. Because the decoder directly outputs a full-resolution image, it cannot generalize to different input or output resolutions at inference time. Supporting a new resolution requires retraining the entire network, making it fundamentally less flexible than our bilateral grid framework, which naturally decouples feature resolution from output resolution through grid slicing. Moreover, the number of parameters required by the DPT model increases as $O(n^2)$ with respect to the output resolution, making the task substantially more difficult than ours, whose parameter count remains fixed.

Overall, in comparison to the DPT head, predicting bilateral grids offers some notable advantages: it enables the use of a compact, low-resolution representation derived from the input image while maintaining compatibility with the original high-resolution image via slicing, as shown in Fig. 7, 8, and 9. This decoupling of input and output resolution is particularly beneficial in memory-constrained environments or when high-resolution outputs are required. For real-time or high-resolution applications (e.g., 1080p or 4K), the bilateral grid head is more practical and efficient, making it especially appealing for tasks such as real-time video processing, mobile deployment, or large-scale inference. Beyond efficiency, another important advantage of the bilateral grid is its ability to learn transformations that adjust global appearance while preserving intrinsic scene shading cues, as demonstrated in Figures 10 and 11 (bottom row), where dark shadows remain dark despite appearance alignment.

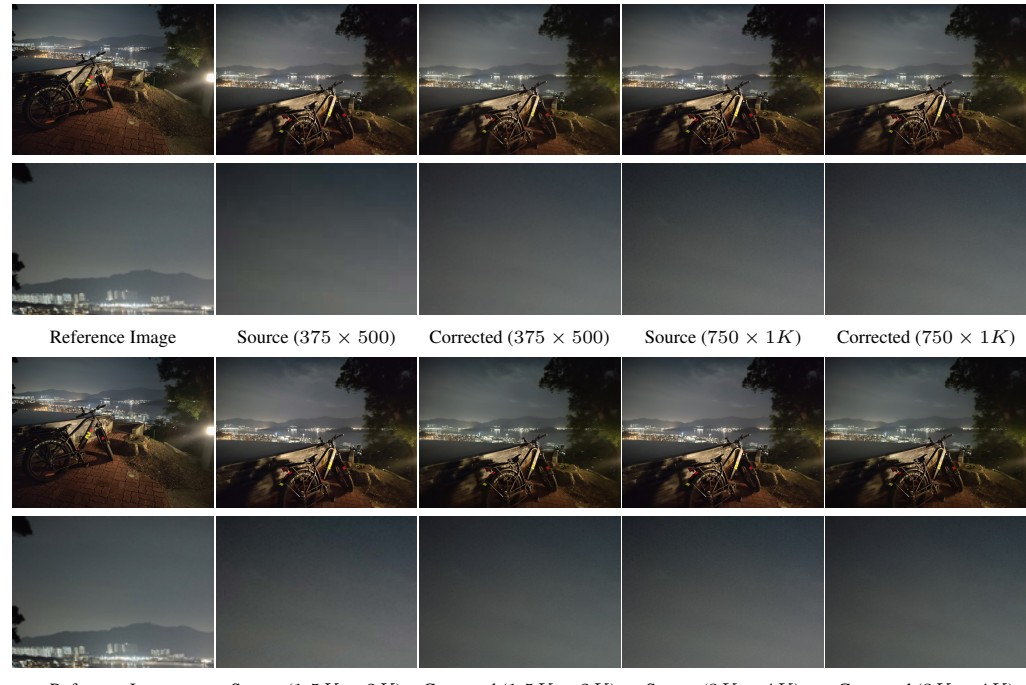

Figure 8: Visualization of our learned bilateral grids applied to low- and high-resolution images, ranging from low-resolution (375 × 500) source and corrected images to their high-resolution (3K × 4K) counterparts. Best viewed when zoomed in.

## A.4 ADDITIONAL QUALITATIVE RESULTS

We provide additional qualitative comparisons on the DL3DV, MipNeRF360-VE, and BilaRF datasets in Fig. 12. Examples of image correction to a given reference frame on the BilaRF dataset are shown in Fig. 13.

## A.5 CONFIDENCE MAP

We provide visualizations in Fig. 18 of the confidence maps predicted by our transformer model alongside the bilateral grids. As described in the main paper, we incorporate confidence map prediction through the Bayesian loss function (Eq. 4) to stabilize training. In Fig. 18, low-confidence regions correspond to areas where reliable color correction is inherently difficult (e.g., saturated highlights or deep shadows), while high-confidence regions align with well-exposed, structurally stable parts of the scene.

## A.6 2D EXPOSURE CORRECTION METHODS

We present qualitative comparisons with 2D exposure correction methods in Fig. 10 and Fig. 11, corresponding to the methods evaluated in Tab. 2 of the main paper. The top row shows input views with significant photometric inconsistencies, including variations in exposure and white balance. While prior methods partially mitigate these issues, they often introduce undesirable artifacts such as color shifts, over-saturation, or unnatural tonal adjustments, especially under challenging lighting conditions. In contrast, our method produces more consistent color and illumination across views, while faithfully preserving fine details and natural scene appearance. Importantly, our approach enables improved multi-view consistency as evidenced by the smoother transitions across viewpoints (e.g., when viewing the corrected frames in Fig. 11 (a) from left to right), which is crucial for downstream 3D reconstruction methods. As shown in Fig. 11 (b), our 3DGS reconstruction of the scene produces a clean, artifact-free wall compared to all other 2D image correction methods. We further conduct an additional experiment comparing 2D image correction methods with ours on the BilaRF dataset, with quantitative results shown in Tab. 9.

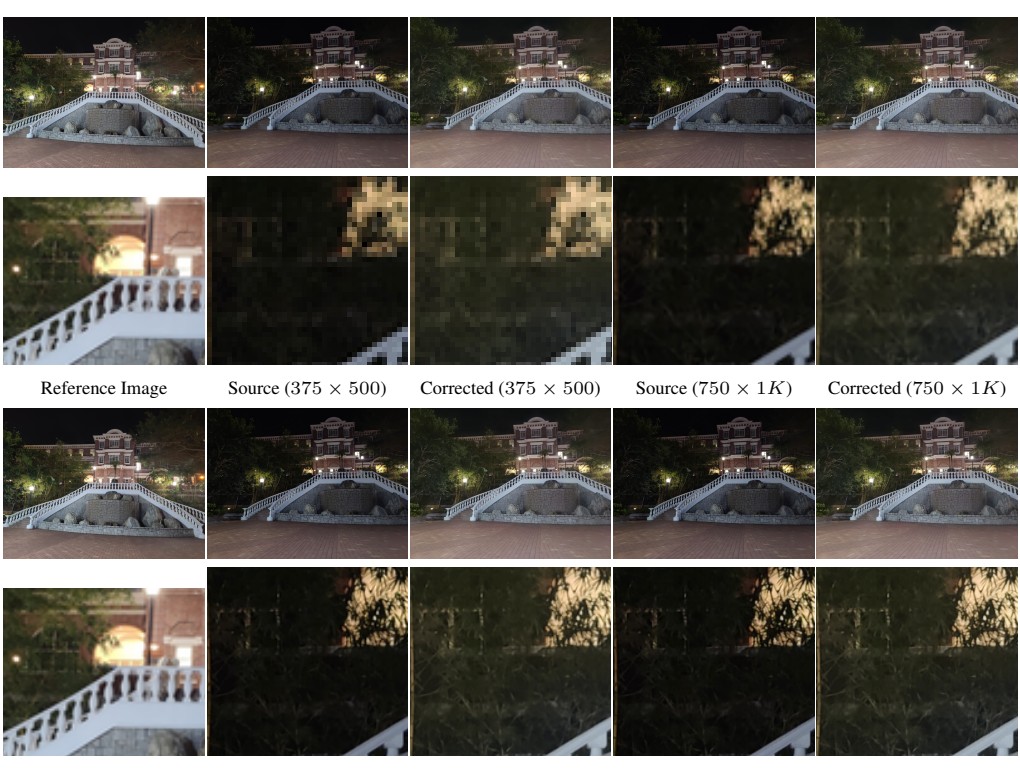

Figure 9: Visualization of our learned bilateral grids applied to low- and high-resolution images, ranging from low-resolution ($375 \times 500$) source and corrected images to their high-resolution ($3K \times 4K$) counterparts. Best viewed when zoomed in.

Table 9: Comparison with 2D image correction methods on the BilaRF Wang et al. (2024a) dataset.

| Comparison with 2D Works | PSNR CC ↑ | SSIM CC ↑ | LPIPS CC ↓ | time ↓ |
|---|---|---|---|---|
| CoTF (Li et al., 2024) | 23.73 | 0.7838 | 0.2743 | 3.01 |
| MSEC (Afifi et al., 2021) | 23.26 | 0.7450 | 0.3615 | 2.42 |
| MSLTNet (Zhou et al., 2024) | 24.29 | 0.7865 | 0.2734 | **1.54** |
| UEC (Cui et al., 2024) | 24.62 | 0.8064 | 0.2488 | 5.36 |
| **Ours** | **25.60** | **0.8240** | **0.2368** | 2.3 |

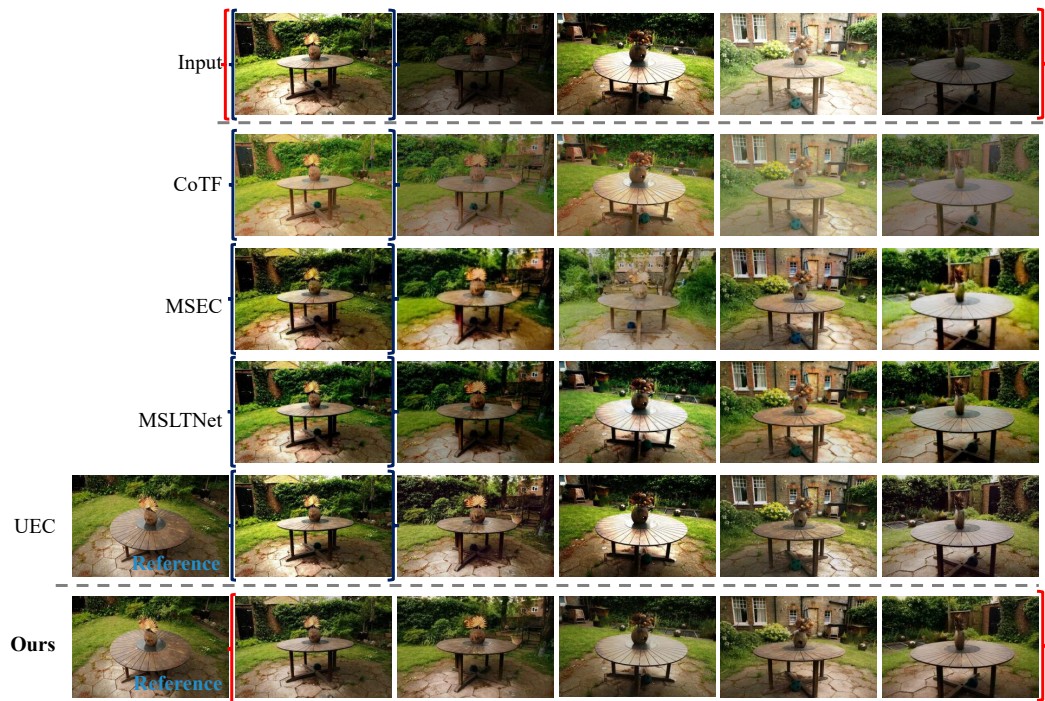

Figure 10: Qualitative comparisons of 2D image correction methods with ours on the garden scene from the MipNeRF360-VE dataset (Cui et al., 2025). All comparison methods (CoTF, MSEC, MSLTNet, and UEC) process frames independently, in contrast to our approach. UEC leverages the same reference frame as ours, selected via our reference-frame selection pipeline.

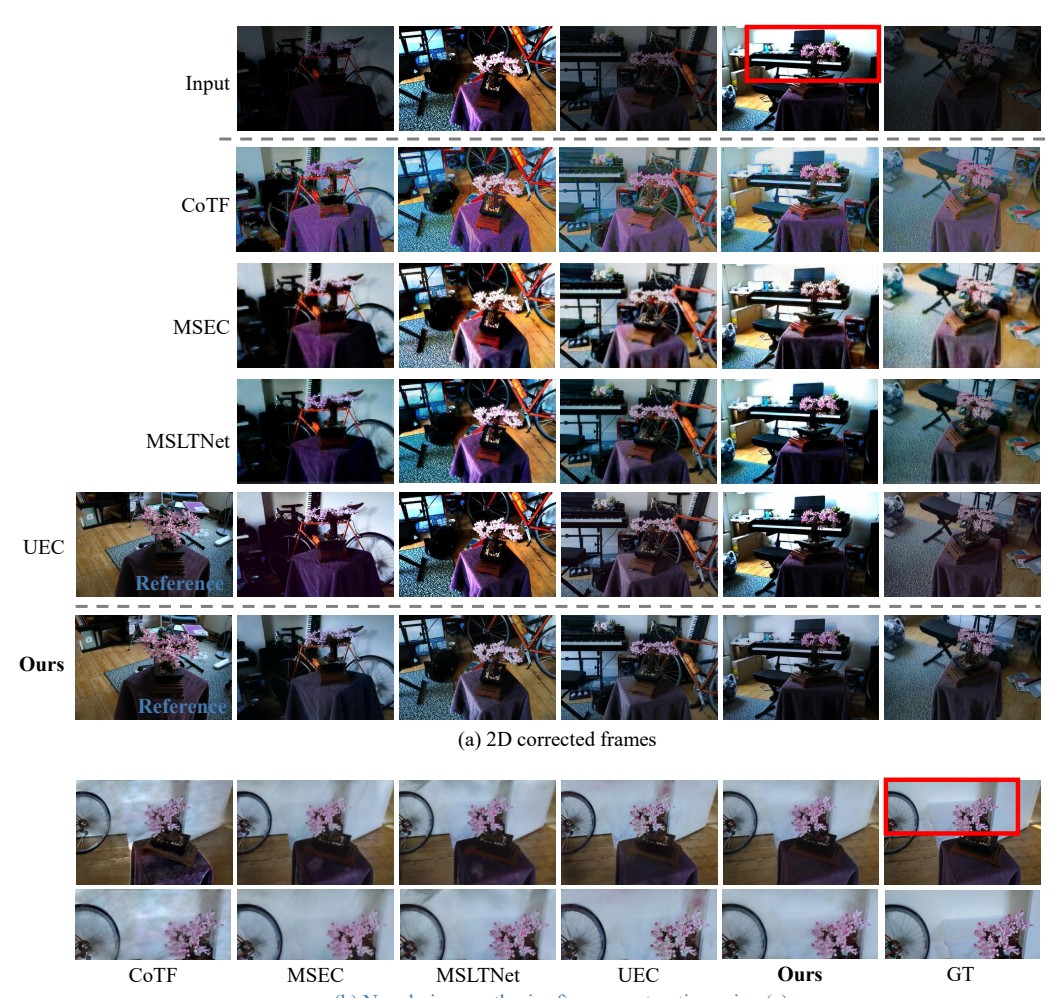

(a) 2D corrected frames

CoTF          MSEC          MSLTNet          UEC          **Ours**          GT

(b) Novel view synthesis after reconstruction using (a)

Figure 11: Qualitative comparisons of 2D image correction methods with ours on the bonsai scene from the MipNeRF360-VE dataset (Cui et al., 2025). All comparison methods (CoTF, MSEC, MSLTNet, and UEC) process frames independently, in contrast to our approach. UEC leverages the same reference frame as ours, selected via our reference-frame selection pipeline. Top (a) shows the corrected 2D frames; bottom (b) shows novel view renders from the resulting 3DGS reconstructions.

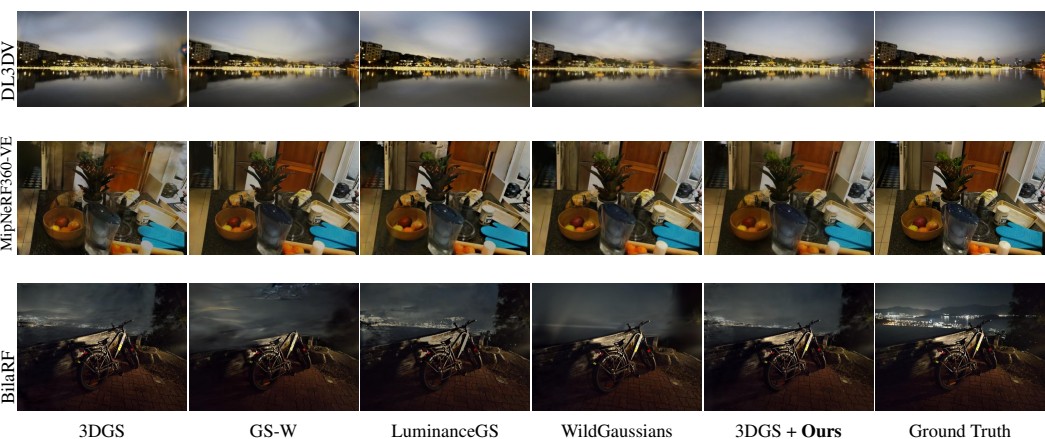

3DGS          GS-W          LuminanceGS          WildGaussians          3DGS + **Ours**          Ground Truth

Figure 12: Qualitative results grouped by dataset: DL3DV, MipNeRF360-VE, and BilaRF.

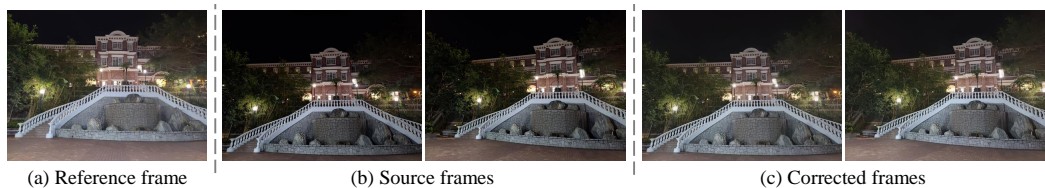

(a) Reference frame     (b) Source frames     (c) Corrected frames

Figure 13: Visualization of image correction by our transformer model. Our model aligns the source frames in (b) to the reference frame in (a), producing the corrected frames shown in (c).

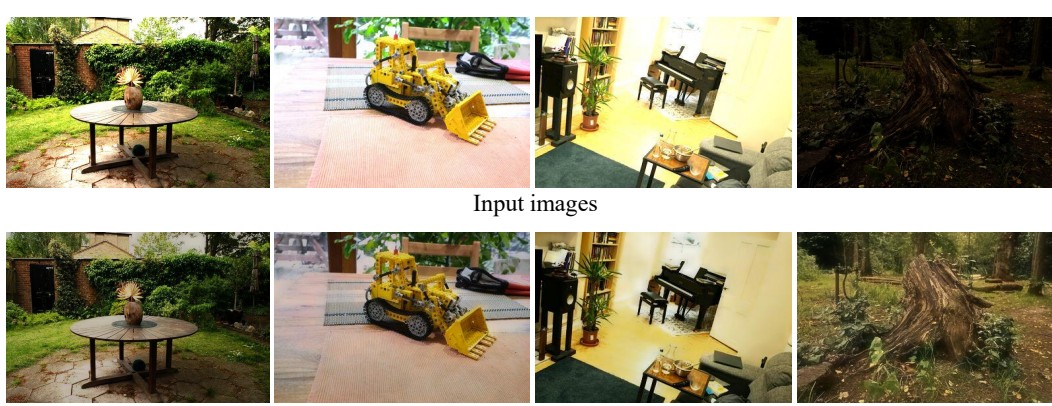

Input images

Corrected images by our model

Figure 14: Visualization of our corrected outputs in the presence of bright light sources and low dynamic-range input images.

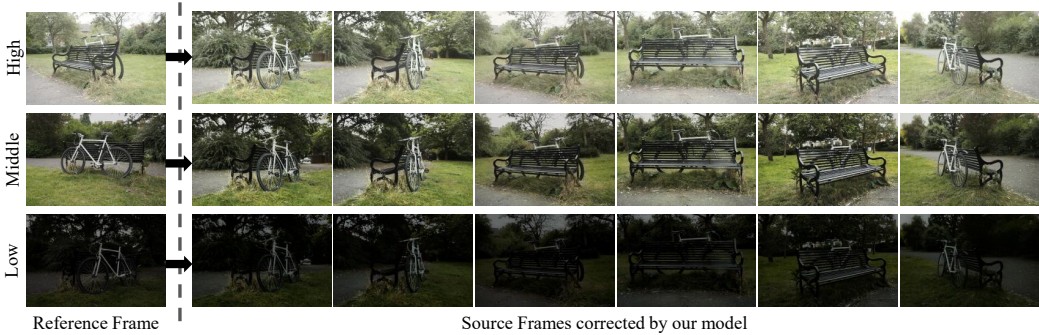

Reference Frame     Source Frames corrected by our model

Figure 15: Visualization of the corrected outputs produced by our model when the reference image is captured under different illumination conditions (low, middle, and high exposure).

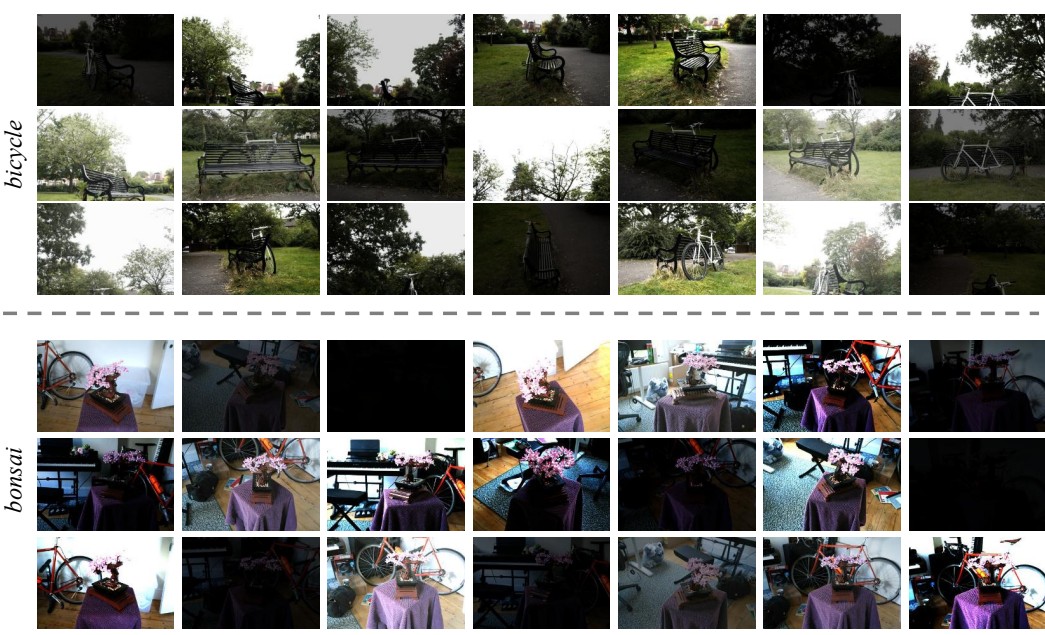

Figure 16: Visualization of a selection of representative input frames from the MipNeRF360-VE dataset.

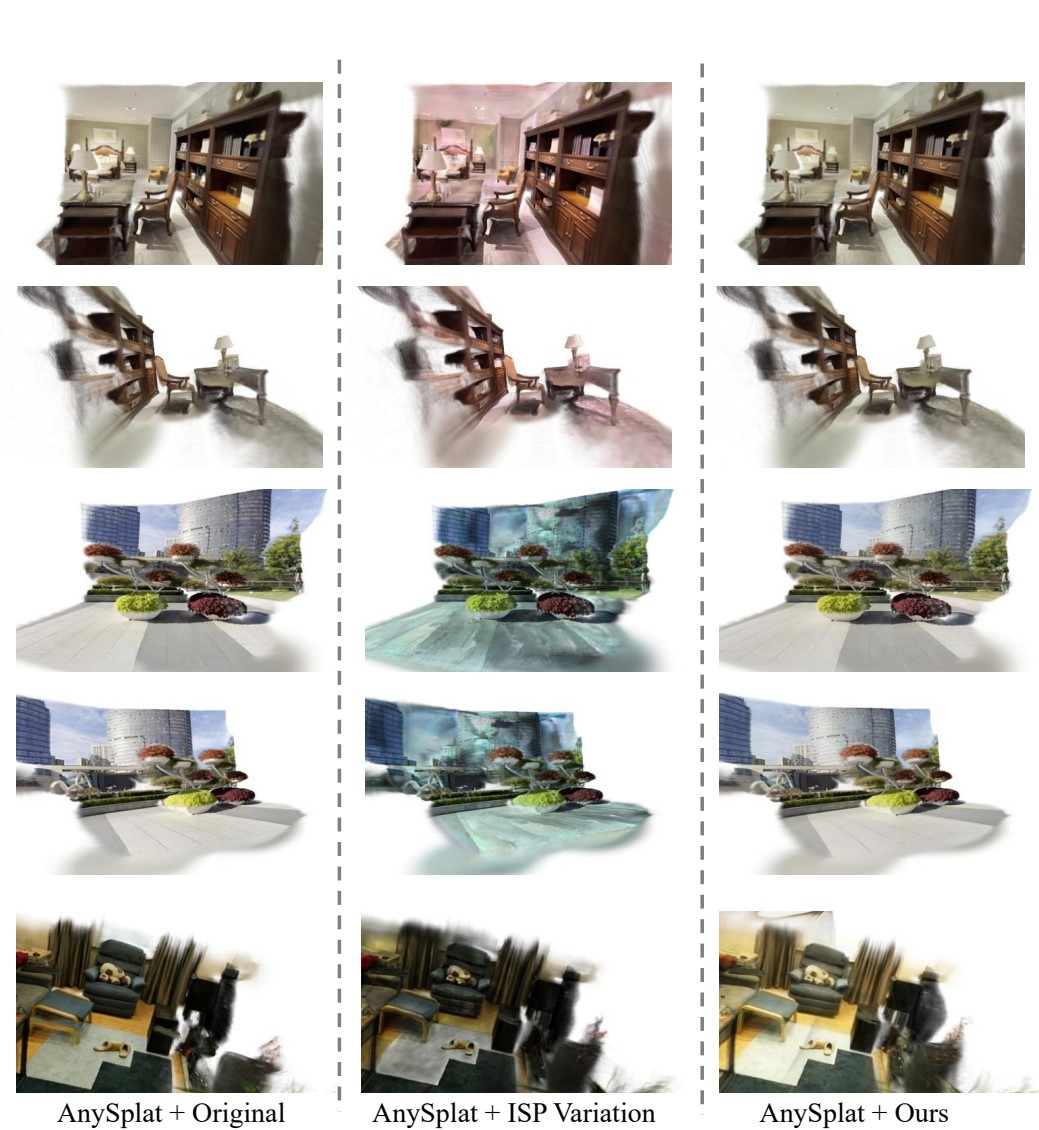

AnySplat + Original      AnySplat + ISP Variation      AnySplat + Ours

Figure 17: Visualization of 3DGS reconstructions using AnySplat for scenes from DL3DV (first four rows) and MipNeRF360 (bottom row). DL3DV input images contain randomized ISP variations, while appearance variations for MipNeRF360 are sourced from the MipNeRF360-VE dataset. From left to right: (1) AnySplat+Original uses the original clean input frames, (2) AnySplat+ISP Variation uses appearance-varying frames, and (3) AnySplat+Ours uses frames corrected by our model.

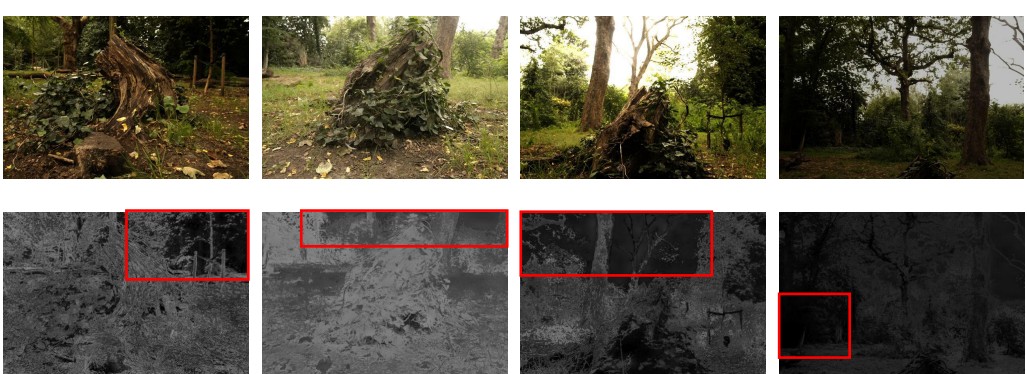

Figure 18: Visualization of confidence maps across input image frames with varying appearance. Darker pixel values indicate lower confidence at the corresponding pixel locations.

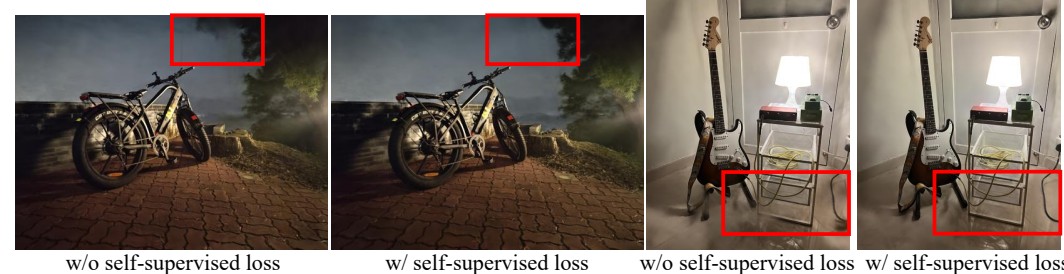

w/o self-supervised loss          w/ self-supervised loss          w/o self-supervised loss    w/ self-supervised loss

Figure 19: Qualitative comparison with and without using AnySplat-based self-supervised loss ($\mathcal{L}_{ss}$).