# OpenReview forum: "CHROMA: Consistent Harmonization of Multi-View Appearance via Bilateral Grid Prediction"
_ICLR.cc/2026/Conference — ICLR 2026 Poster_

### Official Review · Reviewer_bJY1 · 2025-10-26

**Soundness:** 3
**Presentation:** 3
**Contribution:** 3
**Rating:** 8
**Confidence:** 5

**Summary:**

The paper propose CHROMA, to handling novel view synthesis under lightness and ISP variances. Unlike previous methods (such as BiLaRF, Luminance-GS, Wild-Gaussian, etc.), which primarily focus on per-scene optimization and often requiring adjustments to per-scene curves, grids, or other attributes. The proposed method adopts a different solution. it trains a network by synthesizing multi-view data with various ISP degradations. This network includes a transformer encoder, decoder, and grid prediction head. Once trained, the network can serve as an effective multi-view pre-processing tool, restoring multi-view images to standard conditions. This enables the training of a range of multi-view reconstruction methods, such as 3DGS, 2DGS, DashGS, and others. Through the comparison, CHROMA achieve the SOTA results on 3 different datasets.

**Strengths:**

1. This work holds significant practical value. Designing a network capable of harmonizing 3D multi-view images is particularly important, especially given the widespread use of smartphone photography and handheld devices, where camera ISP variance and lightness variance are highly prevalent phenomena.

2. This work eliminates the need for per-scene adjustments on each set of multi-view images, significantly reducing complexity. Although it does not constitute a fully end-to-end network (such as Depth-Splat or MVSGaussian), it already demonstrates substantial practical value. Moreover, experimental results show that CHROMA achieves top-tier performance across all benchmark settings.

3. The paper features an intelligently designed network architecture that effectively reduces inference time and memory consumption (as demonstrated in Table 6 and Table 7), while the writing remains exceptionally fluid throughout.

**Weaknesses:**

1. In summary, this work represents a solid incremental contribution by applying a proven approach, as opposed to formulating a new research problem. Thus I consider its maturity and novelty to be appropriate for a poster, but not sufficient to meet the higher bar for an oral presentation.

2. Regarding the results labeled "+ours" in Table 1, does the reported time include the duration required for the CHROMA network to process the multi-view images? Overall, the reported speed of the proposed method seems almost implausible—for instance, it adds only 0.04s on the BiLaRF dataset. This could easily mislead readers. If such efficiency is indeed achievable, the authors should provide a clear explanation. Additionally, it is important to note whether the inference speed of CHROMA has been evaluated across different GPU hardware.

3. Furthermore, compared to methods like Luminance-GS and GS-W, the authors' approach requires training a network from scratch, which should be explicitly mentioned in the Limitations section.

**Questions:**

While Tables 6 and 7 effectively demonstrate the efficiency gains of the proposed components, it would be equally important to present a comparative analysis of their reconstruction performance. Could the authors provide quantitative results (e.g., PSNR, SSIM, LPIPS) corresponding to the different head designs to offer a more complete picture of the performance-efficiency trade-off?

---

> ### Author Response · Authors · 2025-11-21
>
> We are pleased with reviewer bJY1's positive remarks regarding our work. All comments have been carefully addressed in our revised manuscript, with changes highlighted in green.
>
> **[W2]** We sincerely thank the reviewer for pointing out the potential confusion regarding the reported timings. We would like to kindly mention that the additional runtime reported in the manuscript is 4 seconds, not 0.04 seconds on the BilaRF dataset compared to the original baseline. As stated in the main paper and in the caption of Table 1, all timings reported for +Ours include the full processing time of our model along with the downstream 3D reconstruction. In Section 4.1 of our main paper, we explain that each scene contains approximately 30-70 frames for the BilaRF dataset, and CHROMA requires only 2-3 seconds to process all the frames for each scene. Thus, the observed increase of around 4 seconds is consistent with expectations and reflects a reasonable overhead for the entire multi-view pipeline.
> This efficiency is made possible through the use of FlashAttention and our transformer-based low-resolution bilateral-grid prediction head, which enable fast multi-view processing.
> We also clarify that all inference times were measured on the same mid-range GPU to ensure fairness and consistency across methods. We appreciate the reviewer's observation and clarify this explanation more explicitly in the revised manuscript.
>
> **[W3]** We thank the reviewer for the helpful suggestion. We agree that, unlike methods such as Luminance-GS or GS-W that rely on learning per-view appearance embeddings during scene-specific 3DGS optimization, our approach requires training a dedicated network. We will explicitly mention this in the Limitations section of the revised manuscript.
> At the same time, once trained, our model offers scene-agnostic, feed-forward harmonization and can process hundreds of images in a single pass without any scene-specific optimization. We incorporate this into the revised paper accordingly.
>
>
> **[Q1]** We thank the reviewer for the valuable suggestion. Following the reviewer's recommendation, we replaced our bilateral grid decoder with a dense prediction head (DPT) and trained this variant under the same experimental setup, keeping the total number of parameters comparable to ours. The resulting quantitative comparison is reported in Table R6.
>
> To facilitate stable training, the DPT-based model predicts the residual between the input and ground-truth images, rather than regressing the corrected images directly, following prior work (e.g., StarEnhancer [ICCV 2021]). Despite this adjustment, its reconstruction performance remains significantly below that of our bilateral grid head.
>
> Unlike bilateral grid prediction, the task of dense image-to-image translation is significantly more challenging, requiring substantial architectural engineering and parameter tuning to reach optimal performance.
> Tasks that regress full-resolution images typically rely on components such as multi-scale skip connections, carefully designed feature pyramids, or residual encoders tailored for image reconstruction. Without these specialized structures, the dense prediction head may not fully exploit high-frequency information or photometric cues, which likely explains why its performance remains suboptimal in our controlled comparison.
> Furthermore, a notable disadvantage of the DPT-based model is that the dense prediction head is inherently resolution-fixed.
> Because the decoder directly outputs a full-resolution image, it cannot generalize to different input/output resolutions at inference time. Supporting a new resolution requires retraining the entire network, making it fundamentally less flexible than our bilateral grid framework, which naturally decouples feature resolution from output resolution through grid slicing.
> Additionally, the number of parameters required by the DPT-based model increases by $O(n^2)$ with respect to the output resolution, making the task substantially more difficult than our approach, whose parameter count remains fixed.
>
> *Table R6: Comparison between using bilateral grid decoder and DPT decoder on BilaRF dataset.*
>
> |Method|PSNR CC|SSIM CC|LPIPS CC|
> |---|---|---|---|
> |Dense Prediction Head|24.01|0.7291|0.3882
> |Bilateral Grid Head (Ours)|25.60|0.8240|0.2368

---

### Official Review · Reviewer_b67S · 2025-10-30

**Soundness:** 3
**Presentation:** 3
**Contribution:** 2
**Rating:** 6
**Confidence:** 4

**Summary:**

This paper proposes a novel method for multi-view image harmonization in a feed-forward manner. By employing a VGG-style network architecture to predict a pre-frame bilateral grid, the proposed approach achieves harmonized multi-view images with superior reconstruction quality compared to existing methods.

**Strengths:**

1. The proposed feed-forward framework is novel and well-motivated, effectively improving illumination and color consistency across multiple views and achieving state-of-the-art performance compared to existing methods.
2. This paper is well-written and easy to read.

**Weaknesses:**

1.	The proposed method relies on the assumption that at least one reference view is reliable. However, in real-world scenarios where all input views contain significant artifacts or inconsistencies, the effectiveness of the proposed framework could be severely degraded.
2.	The method models inter-view inconsistencies using a patch-level bilateral grid, which may lead to visible block artifacts. Although a total variation (TV) loss is applied to alleviate this issue, the results are shown only on low-resolution inputs. It remains unclear whether the approach can maintain similar harmonization quality on high-resolution images, where block boundaries could become more noticeable.
3.	The proposed 3D Foundation Model–based self-supervised loss pipeline can improve performance to some extent, but it heavily depends on the chosen pretrained model and introduces considerable training overhead. Moreover, according to the ablation results, the performance gain from this module is marginal, and it even shows a slight decrease in SSIM, suggesting that the inclusion of this component may not be well justified.

**Questions:**

1. How would the method perform if all input views contain artifacts or illumination inconsistencies? Is there any mechanism or potential extension that could improve robustness in such cases?
2. Could the authors provide qualitative or quantitative results on higher-resolution inputs? Since the current experiments are mainly conducted on low-resolution images, it would be helpful to understand how the proposed method performs when applied to higher-resolution data.
3. Could the authors provide more details about the training data scale? For example, how many multi-view scenes or image pairs were used for training?
4. How sensitive is the method to the choice of the pretrained 3D foundation model? If a different foundation model were used, how would it affect the overall performance and stability of the system?

---

> ### Author Response · Authors · 2025-11-21
>
> We thank reviewer b67S for their positive assessment of our work. All feedback has been incorporated into the revised manuscript, with changes highlighted in orange.
>
> **[W1]** We would like to clarify that our method assumes all input views are photometrically inconsistent with each other. Similar to prior appearance-embedding works (e.g., Luminance-GS, BilaRF), we address camera ISP-induced photometric inconsistency, not semantic artifacts or transient corruptions. In our setting, all input views are typically inconsistent (as shown in Figure 16), and the goal is to map them into a single, consistent color space defined by the selected reference frame, regardless of the reference frame's absolute illumination quality. This does not imply that the reference view must correspond to a perfectly illuminated state.
> To further demonstrate the robustness of our approach, we additionally visualize harmonized results obtained using reference frames with different illumination conditions, including cases where the reference does not exhibit ideal lighting. As shown in Figure 15, when the reference frame is either darker or brighter than the source views, our model consistently adapts the corrections to follow the illumination characteristics defined by the chosen reference frame. One possible limitation could be that if there is not any reference image available that is adequate for it to be visualized properly (e.g. all images are underexposed, or no captured image has aesthetically pleasing colors), then we would be unable to improve the harmonization beyond that. Arguably, if that is the case, one could manually edit the image (e.g. by changing the exposure gain, or modifying its color curve) to make it visually appealing, and then harmonize all input images to that. However, it is unlikely in practice that no usable reference frame would be captured.
> Furthermore, artifact-heavy inputs, such as transient objects or motion blur, represent a fundamentally different problem from ISP-related appearance harmonization. We would like to clarify that handling such artifacts is outside the scope of the current work. Extending our framework to fully in-the-wild scenarios containing transient objects, as we mentioned in future work, would require an additional semantic-aware module capable of identifying and isolating such effects.
>
> **[W2]** We agree that patch-based bilateral grid methods may, in principle, suffer from block boundary artifacts. This limitation is well known in prior bilateral-grid approaches ([HDRNet, SIGGRAPH 2017], [Decolorful-Net, CVPR 2024]), which mitigate it by combining spatially local features with global feature cues when predicting the grid. In contrast to these CNN-based designs, our transformer architecture naturally aggregates global context through attention layers, enabling the predicted bilateral grids to remain smooth across patch boundaries.
> As a result, we do not observe block artifacts even at high resolutions. While a high-resolution example is already included in Figure 7, we additionally provide further visualizations in Figures 8 and 9 in the revised manuscript. In particular, Figure 8 includes a zoomed-in view of the dark sky region exhibiting a vignette effect, where block artifacts commonly arise. The corrected results clearly show that no block artifacts are introduced by our model.
> We appreciate the reviewer's comment, as this highlights an additional advantage of our design. We will explicitly mention in the revised manuscript that the transformer-based architecture effectively suppresses block-boundary issues while enabling high-resolution harmonization.

---

> ### Author Response · Authors · 2025-11-21
>
> **[W3] and [Q4]**
> We thank the reviewer for raising these points. First, we would like to clarify that the performance gain from our 3D foundation model-based self-supervised loss is meaningful. A +0.4 dB PSNRCC improvement is considered significant in reconstruction settings where the baseline is already strong and the remaining inconsistencies are subtle. We additionally provide novel-view visualizations in Figure 18 showing the qualitative benefits of using the self-supervised loss. Also, the self-supervised loss ultimately serves as a regularizer rather than a dominant supervisory signal. As shown in Table R1, the L1 variant (more sensitive to low-level artifacts from AnySplat) provides little improvement, while the VGG-based perceptual loss offers a more stable and informative signal. This shows that CHROMA is not overly sensitive to the fine-grained photometric biases of the foundation model.
>
> Regarding the dependence on the pretrained 3D foundation model, we acknowledge that our approach relies on the chosen model. At the time of submission, AnySplat was the only feed-forward 3D foundation model that is publicly available, being capable of taking only images as input (i.e., pose-free) and predicting both camera poses and Gaussian primitives, making it the only model compatible with our self-supervised pipeline. AnySplat is trained at scale using the VGGT [CVPR 2025] backbone and provides sufficiently reliable multi-view predictions for our consistency objective. To the best of our knowledge, it remains the only model with this capability, aside from FLARE [CVPR 2025], which reports substantially lower performance in Table 1 of the AnySplat paper.
> In addition, the way our model utilizes the foundation model requires a modest amount of architectural alignment. As shown in Figure 3, CHROMA separates per-view features so that only the corrected source views are passed to the Gaussian decoder while the predicted reference pose is used for rendering. This required adapting AnySplat's interface to expose pose prediction, Gaussian decoding, and differentiable rendering.
>
> In the revised manuscript, we will make clear that we choose AnySplat as the 3D foundation model, and that the claims of our work are related to this specific choice. We will also clarify the rationale for using AnySplat and the nature of this dependence in the revised manuscript. If the reviewer is aware of a more suitable feed-forward 3D foundation model that provides stable pose estimation, Gaussian prediction that is separable to each view, and differentiable rendering in a fully feed-forward manner, we would greatly appreciate the recommendation. To the best of our knowledge, AnySplat remains the strongest available option that satisfies all these requirements.
>
> *Table R1: Ablation of using L1 loss vs VGG loss for self-supervision.*
>
> |Method|PSNR CC|SSIM CC|LPIPS CC|
> |---|---|---|---|
> |w/o $\mathcal{L}_{ss}$|25.21|0.8245|0.2375
> |L1-based $\mathcal{L}_{ss}$|25.28|0.8227|0.2377
> |VGG-based $\mathcal{L}_{ss}$ (Ours)|25.60|0.8240|0.2368
>
> **[Q2]** We appreciate the reviewer's suggestion. Although our method operates in a resolution-independent manner, we agree that showing explicit results on high-resolution inputs is valuable. We have already included a high-resolution analysis in Table 7, which demonstrates that our bilateral grid formulation scales efficiently with increasing resolution.
> To further address the reviewer's concern, we include additional qualitative high-resolution results in Figures 8 and 9. For quantitative evaluation, Table R5 provides a comparison between 3DGS and 3DGS + Ours using high-resolution inputs. These results consistently show that our harmonization remains stable and effective as resolution increases. It is also worth mentioning that our model does not exhibit significant computational increase along with the resolution increase because we predict bilateral grids instead of images, as shown in Tables 6 and 7.
>
> *Table R5: Model performance with different input resolutions, comparing 3DGS and 3DGS+Ours on BilaRF dataset.*
>
> |Ours+3DGS| $375\times 500$ | $750\times 1K$ | $1.5K\times 2K$ | $3K\times 4K$|
> |---|---|---|---|---
> |Inference time|2.3s|2.88s|5.46s|13.14s
> |PSNR CC $\uparrow$|25.60 (+5.65\%)|25.28 (+5.91\%)|25.35 (+10.21\%)|25.01 (+6.79\%)
> |SSIM CC $\uparrow$|0.8240 (+2.76\%)|0.8207 (+4.94\%)|0.8123 (+7.90\%)|0.7906 (+3.96\%)
> |LPIPS CC $\downarrow$|0.2368 (-8.71\%)|0.2636 (-9.45\%)|0.2757 (-13.84\%)|0.3062 (-13.91\%)

---

> ### Author Response · Authors · 2025-11-21
>
> **[Q3]**
> Our training set combines both paired synthetic data and unpaired real-world data, as described in Section 3.4. In detail, for paired data, we use 10K multi-view scenes from the DL3DV dataset, where each scene contains roughly $300$ frames. As discussed in Section 3.4, we further apply random realistic ISP variations at each iteration, thus yielding a large data augmentation on top of the original DL3DV dataset. For unpaired real-world data, we additionally use 8K recorded objects from the WildRGB-D dataset, where each scene also contains roughly $300$ frames. This dataset exhibits real ISP-induced inconsistencies while capturing the objects from 360 degrees. We include these dataset statistics in the revised manuscript for clarity.

---

### Official Review · Reviewer_UcJY · 2025-10-31

**Soundness:** 3
**Presentation:** 3
**Contribution:** 2
**Rating:** 4
**Confidence:** 5

**Summary:**

1. This paper studies the task of multi-view appearance harmonization. The goal is to make images of the same scene from different viewpoints consistent in color, exposure, and white balance, which helps improve 3D reconstruction and novel view synthesis quality.

2. Previous works did not focus on feed-forward harmonization. This paper is the first to explore a feed-forward solution for this task.

3. The authors propose a general transformer that predicts a 3D bilateral grid for each view to achieve consistent color mapping. The bilateral grid provides spatial adaptivity and edge preservation, making it effective for modeling ISP-related variations.

4. To train the model, the authors use DL3DV with synthetic ISP augmentations and introduce a rendering consistency loss based on a 3D reconstruction model.

**Strengths:**

1. The method is technically sound. The bilateral grid is well suited for spatially adaptive and edge-preserving color correction.

2. The experiments are thorough and well validated.

**Weaknesses:**

1. The authors train the model on DL3DV with handcrafted ISP variations and also test on DL3DV with the same synthetic setup. This limits the significance of the results, since a feed-forward model will naturally perform well under similar handcrafted conditions.

2. In Table 1, the improvement over Luminance-GS on the LOM and BilaRF datasets appears small.

**Questions:**

Please address the issues listed in the weaknesses section.

---

> ### Author Response · Authors · 2025-11-21
>
> **[W1]** We agree with the reviewer that evaluating solely on DL3DV with similar synthetic ISP variation would not sufficiently demonstrate the generality of our model. This is the reason why we conduct comprehensive evaluations on two additional public datasets, MipNeRF360-VE and BilaRF, both of which lie outside the distribution of our synthetic DL3DV augmentations. MipNeRF360-VE contains random exposure and gamma variation, and BilaRF contains real-world variation with uncontrolled camera ISP behavior. We kindly note that we have revised the dataset name from LOM to MipNeRF360-VE in Table 1 for clarity to address a comment from another reviewer. We confirm that this naming correction does not affect the experimental setup or the datasets used.
> As shown in Table 1 and Figure 4 of the main paper, our method consistently achieves the best performance across all three datasets, including these out-of-distribution benchmarks with very different exposure levels, illumination conditions, and ISP characteristics. This strongly suggests that our model does not simply overfit to handcrafted ISP perturbations, but instead learns a representation that generalizes well across a broad spectrum of real camera pipelines and appearance inconsistencies. We would like to note to the reviewer that Section 4 of the main paper discusses this and shows that our model outperforms existing methods on all those datasets, despite the distinct characteristics among them. To emphasize the performance on real-world variations, we present the main quantitative results on the BilaRF dataset in Table R4.
>
> *Table R4 Quantitative results on BilaRF dataset (from Table 1 in the main paper).*
>
> | Method | PSNR CC | SSIM CC | LPIPS CC | Time |
> |---|---:|---:|---:|---:|
> | 2DGS | 23.43 | 0.746 | 0.308 | 13m 26s |
> | 3DGS | 24.23 | 0.8019 | 0.2594 | 3m 43s |
> | DashGS | 24.34 | 0.7880 | 0.2607 | 3m 46s |
> | WildGaussians | 23.19 | 0.7424 | 0.3121 | 1h 58m 06s |
> | GS-W | 24.94 | 0.8056 | 0.2764 | 40m 34s |
> | Luminance-GS | 23.41 | 0.7931 | 0.2750 | 18m 40s |
> | 2DGS-4DBAG | 24.80 | 0.773 | 0.273 | - |
> | 3DGS-4DBAG | 24.90 | 0.774 | 0.256 | - |
> | **2DGS + Ours** | **25.27** | **0.8147** | **0.2499** | 13m 30s |
> | **3DGS + Ours** | **25.60** | **0.8240** | **0.2368** | 3m 48s |
> | **DashGS + Ours** | **26.25** | **0.8356** | **0.2158** | 3m 50s |
>
>
> **[W2]** We believe there may have been a misunderstanding regarding the magnitude of improvement produced by our method. The relevant metrics for comparison are PSNR**CC**, SSIM**CC**, LPIPS**CC**, which refer to the color-corrected metrics also used in BilaRF [BilaRF, SIGGRAPH 2024]. Since different appearance-correction methods may produce outputs in different global color spaces, raw metrics can be misleading. Metrics after applying per-channel global affine transformation (CC) evaluate the reconstruction after aligning global affine color transforms and are therefore a more reliable measure of geometric and structural fidelity. We kindly note that, as presented in Table 1, our method shows substantial gains over the comparison methods on the Luminance-GS MipNeRF360-VE dataset (previously LOM dataset): +3.13 dB (+13.6\%) improvement (23.12 vs 26.25 with 3DGS) and on the BilaRF dataset: +2.19 dB (+9.35\%) improvement (23.41 vs 25.60 with 3DGS) in PSNRCC.
> These gains are consistent and significant across both out-of-distribution datasets, which also relates to the reviewer's concern above (W1).

---

### Official Review · Reviewer_u8mx · 2025-11-03

**Soundness:** 3
**Presentation:** 3
**Contribution:** 3
**Rating:** 6
**Confidence:** 3

**Summary:**

The paper presents a feed-forward approach for appearance adjustments in order to achieve photometric consistency across multiple views that can then improve the accuracy of downstream 3D reconstruction tasks. The proposed method learns spatially varying bilateral grids for edge-aware photometric adjustments. The paper additionally includes a methodology for selecting an appropriate reference frame for photometric alignment, and a self-supervised training paradigm by optimizing over the downstream 3D reconstruction achieved from consistent novel views.

**Strengths:**

The paper is well written and theoretically detailed. The overall evaluation presented in the paper is mostly thorough and shows favourable results from the proposed approach. It is particularly interesting to see considerable improvement in 3D reconstruction (Fig 4).

The theoretical contributions in the paper are more or less incremental when viewed in the context of existing literature, however, they seem to be put together in a coherent manner in the proposed approach. Bilateral grids for per-image appearance editing while enforcing consistency using self and cross attention are the main contributions in my opinion.

Selecting the correct reference frame is quintessential for the success of this approach, and it is good to note that the authors have discussed this at length, considering both extrema of high and low luminosity and avoiding them.

Evaluating the method across three datasets covering synthetic, exposure-stacking and wild scenes is commendable and the proposed method shows favorable results.

**Weaknesses:**

One of the main challenges in photometric consistency across multiple views is in being able to handle specularities - which is where most methods fail remarkably. While specular surfaces may be significantly challenging to address, varying specularities across views from bright light sources and low dynamic range of sensors is certainly applicable to the proposed approach. I would expect the approach to at least discuss and/or present the performance on some examples.

The image in column 3 of Figure 10 is particularly confusing. It seems as if a uniform scaling down of the luminance is applied here without any contrast stretching in the input image. However, all the demonstrated methods show varying levels of contrast stretching, including the proposed approach. I presume there is a mismatch between input and generated images, in my opinion.

There are some examples where the qualitative improvement is not immediately apparent, if at all, in the figures presented in Figure 11.

Figure 4 is where the true impact on geometry is visible on using the proposed approach. If one were to compare 3DGS versus 3DGS+Ours for the Bicycle scene, the fine details in the bicycle and the bench are clearly visible in the proposed method, however, it is not immediately apparent why this may be achieved by photometric consistency across views alone, particularly because of the lack of significant photometric variation in the direction of view for these parts of the scene.

As is the case with several self-supervised approaches, there is an inherent assumption here that the output of the 3D foundation model (Anysplat) is photometrically reliable. Anysplat itself relies on scene visibility, textural details and implicit sparse depth across views. Some analysis on the self-reinforcement caused by this assumption will be useful to the reader.

The confidence aware prediction mechanism is appreciated, however, I believe it is also important to illustrate some of the per-pixel confidence maps (l.242) of a few views from the same scene to actually decipher the impact of the proposed method. There is no mention of this in the paper.

Given the efficiency and easy-integration proposed in the paper, more discussion on temporal consistency may be useful for dynamic scenes and may result in a wider contribution (extending the discussion about Table 2)

Affine warping in a patch based manner (as proposed in the paper) brings with it the likelihood of washing out legitimate scene properties such as shadows - this may be useful from the standpoint of geometric reconstruction, but not so much in terms of photo-realistic view synthesis. I would expect the authors to mention or discuss this point, even if briefly.

2D-only methods like UEC and MSEC are more relevant for comparison in terms of compute time. I would urge the authors to provide that comparison.

**Questions:**

See points under the Weakness section

---

> ### Author Response · Authors · 2025-11-21
>
> We are pleased with reviewer u8mx's positive remarks regarding our work.
> We have addressed all points raised by reviewer u8mx, with changes to the revised manuscript highlighted in blue.
>
> **[W1]** The datasets used in our experiments contain some specularity inconsistencies the reviewer mentions, particularly those arising from bright light sources and low dynamic range (LDR) sensor responses. As shown by the input frames in Figure 16 of the revised manuscript, such view-dependent intensity spikes and localized highlight shifts are naturally present in certain scenes.
> While it is true that severely clipped highlights cannot be fully reconstructed, since the underlying information is lost during the sensor capture, we observe that our model successfully stabilizes the tonal inconsistencies across views by leveraging multi-frame processing in a single step. This effect can be seen in the outputs shown in Figures 10 and 11, where regions affected by clipping remain photometrically consistent in terms of tone across the corrected views, despite their challenging nature. Since our model considers all frames jointly, it is able to harmonize appearance based on information available in other views, reducing the visual impact of specularity-induced fluctuations.
> To address the reviewer's request directly, we additionally include Figure 14 in the revised manuscript. This example further confirms that while our bilateral grid-based model cannot reconstruct information that is fundamentally lost (e.g., saturated pixels), it nevertheless provides stable corrections in the presence of varying specularities induced by illumination and sensor limitations.
>
> **[W2]** We appreciate the reviewer's careful inspection of Figure 10 and clarify the setup to avoid any misunderstanding. The top row in Figure 10 shows the input frames for each correction method (methods listed on the left). Our method, being a multi-view transformer, uses all frames as input along with a selected reference frame. Note that UEC uses the same reference, which we now visualize alongside the corrected outputs in Figure 10 of the revised manuscript.
>
> The example in Figure 10 comes from the MipNeRF360-VE validation dataset introduced in Luminance-GS (CVPR 2025), where all inputs contain both luminance and contrast variations. For visualization, we choose five input frames, corresponding to the five columns of Figure 10, and each method's corrected output for that specific frame appears in the matching column.
>
> All methods evaluated in this figure are trained to handle exposure and gamma variations, so contrast changes in the corrected outputs are expected. The reviewer is correct that the input in column 3 shows reduced luminance but minimal contrast change. However, the varying degrees of contrast stretching observed in the outputs are a result of each method's correction behavior, not a mismatch. In our case, the method harmonizes each frame to the reference, which in this example has noticeably higher contrast than the column 3 input, explaining the apparent increase in contrast.
>
> Therefore, we confirm there is no mismatch between inputs and outputs. That said, we acknowledge that the column 3 example may unintentionally cause confusion. In the revised manuscript, we have replaced this particular sample with a clearer one and also include the corresponding reference frame to improve interpretability.
>
> **[W3]** We would like to clarify that Figure 11 (a) does not show 3D reconstruction results. Instead, it presents 2D corrected frames produced by each method before 3DGS reconstruction. The purpose of this figure is to highlight temporal and cross-frame photometric consistency under large exposure and gamma variations. In this setting, inconsistencies are most evident when comparing frames from left to right, for example, noticeable shifts in wall color or overall brightness. Our method's multi-view formulation significantly reduces these variations compared to the other methods, resulting in substantially more consistent appearance across frames.
>
> To better illustrate the downstream impact and qualitative improvement, we have added Figure 11 (b) to the revised manuscript, visualizing the resulting 3DGS reconstructions derived from the corrected frames in Figure 11 (a). Here we observe significantly more artifacts in the novel view renders from the other methods compared to our result which is much cleaner. These results show that improved frame consistency leads to more stable and coherent 3D reconstructions, reinforcing the benefit of our approach.

---

> ### Author Response · Authors · 2025-11-21
>
> **[W4]** Upon reviewing results for the Bicycle scene in Figure 4, we note that although we provided the right citation and description of the dataset (Cui et al., CVPR 2025), we used the LOM dataset denomination incorrectly, and have therefore updated the revised manuscript to refer to the MipNeRF360-Varying Exposure (MipNeRF360-VE) dataset instead, which is the original dataset denomination used in Luminance-GS (Cui et al., CVPR 2025). We apologize for the confusion caused by this, and we confirm that this correction does not affect the experimental setup or the datasets used. All changes have been fully reflected in the revised manuscript.
>
> The MipNeRF360-VE dataset indeed exhibits substantial exposure variations across all views. These variations are observable even on the bicycle and bench surfaces in the Bicycle scene, consistent with the dataset description provided in Luminance-GS, where this testing set was first introduced. To visualize the input of the dataset, we refer the reviewer to Figure 3, where we show the input-frame visualizations (Frame 1, 2, and 3 in the reference frame selection section), for this very same Bicycle scene example. We showcase more examples of the inputs in the newly added Figure 16 of the manuscript.
>
> When these variations are not corrected prior to reconstruction or during reconstruction, they interfere with the multi-view photometric consistency required by 3DGS, and consequently hinder stable geometry optimization. By harmonizing these variations before 3DGS optimization, our method yields the fine details in novel views as the reviewer mentioned in Figure 4, where we have additionally included example input images with variation (from left to right).
>
> **[W5]** We thank the reviewer for highlighting this point. AnySplat is explicitly trained to enforce photometric consistency through its Gaussian rendering head. Photometric loss is computed between the rendering of Gaussians and the training views (i.e. forcing the model to converge to a photometrically consistent solution, as it does not have expressivity to model strong, per-input-view appearance variations), thus we argue that it is well aligned with our experimental setting. This is also reflected in the ablation in Table 3 and Figure 19, where incorporating our self-supervised loss yields a clear improvement over the purely supervised (i.e. with synthetic data) variant. Furthermore, as shown in Figure 17, AnySplat produces accurate novel-view predictions when the inputs are photometrically consistent, while its performance degrades under strong photometric inconsistencies. We believe this is evidence that the 3DGS reconstruction from AnySplat is a good pretext task to provide guidance to improve photometric consistency where no ground-truth images can be obtained. Our method intentionally leverages this behavior to reinforce multi-view consistency across frames.
>
> Nevertheless, we agree that AnySplat's predictions depend on factors such as scene visibility and texture. Therefore, we have carefully designed our training setup to mitigate potential self-reinforcement issues. We use the predicted Gaussian opacity (alpha) to mask out unreliable regions and apply the self-supervised loss only to well-reconstructed areas. Additionally, to prevent the model from overfitting to minor detail loss that may inevitably occur in AnySplat's renderings, we adopt a VGG-based perceptual loss, which is more robust to low-level photometric noise or misalignment than a pixel-wise L1 loss. As shown in Table R1, the VGG-based loss provides a more stable and informative consistency signal, leading to noticeably stronger performance compared to the L1 variant. We have included this analysis in the revised manuscript.
>
> *Table R1: Ablation of using L1 loss vs VGG loss for self-supervision.*
>
> |Method|PSNR CC|SSIM CC|LPIPS CC|
> |---|---|---|---|
> |w/o $\mathcal{L}_{ss}$|25.21|0.8245|0.2375
> |L1-based $\mathcal{L}_{ss}$|25.28|0.8227|0.2377
> |VGG-based $\mathcal{L}_{ss}$ (Ours)|25.60|0.8240|0.2368
>
> **[W6]**
> We appreciate the reviewer for emphasizing the importance of visualizing the predicted confidence maps. We include visualization in Figure 18 of the revised manuscript, where low-confidence regions correspond to areas where reliable color correction is inherently difficult (e.g., saturated highlights or deep shadows), and high-confidence regions align with well-exposed, structurally stable parts of the scene.

---

> ### Author Response · Authors · 2025-11-21
>
> **[W7]**
> We appreciate the reviewer's suggestion regarding the extension to dynamic scenes. Currently, our scope has been on static scene reconstruction only. As shown in Table 2, we examine temporal consistency across views, which reinforces our main goal. We agree that handling dynamic scenes would be interesting future work. This could be achieved by integrating existing representations for explicit dynamics modeling, such as Ex4DGS [NeurIPS 2024] which distinguishes between approximately static background regions and truly moving objects. However, this would require substantial modifications to our proposed method, i.e. moving to large-scale dynamic datasets as well as changes to the self-supervised rendering loss as AnySplat cannot handle dynamic objects.
>
> **[W8]**
> Our bilateral grid-based correction does not remove or override legitimate scene properties such as shadows. The bilateral grid predicts smooth, low-resolution affine color transforms that are intentionally limited to modeling low-frequency spatially-varying ISP-induced variations (e.g., exposure, white balance, tone mapping), rather than high-frequency, illumination-dependent cues.
> Because the grid is defined at a coarse spatial resolution with a 1D luminance guidance dimension, it cannot introduce or erase high-frequency structures, including shadow boundaries, shading gradients, or object contours. These components remain intact in the corrected images.
> Moreover, our multi-view transformer model enforces inter-frame consistency while our reference-frame selection strategy selects a semantically rich reference frame with balanced exposure, making cases where shadows would be washed out extremely unlikely in practice. As shown in Figures 10 and 11 (bottom row), the learned transformations align the overall appearance of the frames without interfering with scene-intrinsic shading cues (dark shadows remain dark).
>
>
> **[W9]**
> We appreciate the reviewer's suggestion and agree that compute-time comparison is important when evaluating against 2D-only exposure-correction baselines. To address this, we have conducted additional experiments measuring the image processing time on both the MipNeRF360-VE and BilaRF datasets, combined with 3DGS reconstruction, in Tables R2 and R3.
> Since MipNeRF360-VE scenes contain over 200 frames and our model uses a transformer architecture that processes all views jointly, CHROMA has a few additional seconds of overhead compared to CNN-based 2D methods that operate per-frame. However, this overhead is small relative to the overall reconstruction time of 3DGS, which takes few minutes, meaning the additional cost does not meaningfully affect the total pipeline runtime.
> More importantly, our method consistently achieves the best performance across both datasets containing various illumination variations, with particularly large gains (nearly +1 dB PSNRCC) on the BilaRF dataset.
>
> *Table R2 Compute time comparison between 2D-only methods+3DGS and ours+3DGS on MipNeRF360-VE dataset.*
>
> |Method|CoTF|MSEC|MSLTNet|UEC|Ours
> |---|---|---|---|---|---
> |Inference Time |3.46|7.79|3.12|7.64|8.59
> |PSNRCC|23.15|23.98|24.56|25.98|**26.25**
> |SSIMCC|0.7573|0.7169|0.7697|0.8143|**0.8149**
> |LPIPSCC|0.2698|0.3359|0.2697|**0.2416**|0.2428
>
> *Table R3 Compute time comparison between 2D-only methods+3DGS and ours+3DGS on BilaRF dataset.*
>
> |Method|CoTF|MSEC|MSLTNet|UEC|Ours
> |---|---|---|---|---|---
> |Inference Time |3.01|2.42|1.54|5.36|2.3
> |PSNRCC|23.73|23.26|24.29|24.62|**25.60**
> |SSIMCC|0.7838|0.7450|0.7865|0.8064|**0.8240**
> |LPIPSCC|0.2743|0.3615|0.2734|0.2488|**0.2368**

---

### Comment · Area_Chair_cuMk · 2025-11-27

Dear Reviewers,

As we enter the discussion phase, I strongly encourage you to read the authors' rebuttal carefully and acknowledge their effort. Silence is the worst outcome for an author. Even if the rebuttal does not change your final rating, a brief response explaining why the concerns remain unaddressed is crucial for a fair process. Please help us make an informed decision by engaging in a constructive dialogue.

AC

---

> ### Author Response · Authors · 2025-12-03
>
> Dear Area Chair, we would like to express our sincere gratitude to the reviewers for their time, constructive feedback, and service, and to you for coordinating the review process for our submission.
>
> We would also like to inform you that, although we carefully addressed every comment during the rebuttal period, no further replies or follow-up questions were made by the reviewers in the subsequent seven days. Unfortunately, after this period, an OpenReview system leakage occurred, preventing reviewers from submitting post-rebuttal scores, even though we had provided detailed, point-by-point responses in the comments.
>
> For your convenience, we include a summary of our rebuttal activity, outlining how we addressed each reviewer's concerns, as well as the clarifications and revisions incorporated into the manuscript. The key points addressed in the revised version are as follows:
>
> -----------------------------------------------------------------------
>
> **Reviewer u8mx:** We added analysis on specularity-related inconsistencies (**[Fig. 14, Lines 528-530]**), clarified the setups of Fig. 10-11 with additional examples, updated the dataset denomination of LOM to MipNeRF360-VE, and included input images in Fig. 4. We analyzed the AnySplat-based self-supervised loss (**[Tab. 3, Lines 309-311, Lines 503-507, Fig. 17, 19]**), visualized the confidence maps (**[Fig. 18]**), discussed the static-scene scope and shadow preservation (**[Lines 915-917]**), and provided compute-time comparisons against 2D-only baselines with further ablations (**[Tab. 2, 9, Lines 430-431, 481-483]**).
>
> **Reviewer UcJY:** We emphasized that we train on DL3DV with synthetic ISP variation but evaluate on out-of-distribution MipNeRF360-VE and BilaRF datasets. We clarified that the relevant PSNR CC, SSIM CC, and LPIPS CC metrics show substantial improvements **(e.g., +3.13 dB (+13.6\%) and +2.19 dB (+9.35\%) PSNR CC)** over Luminance-GS on those datasets (**[Tab. 1]**). Based on the reviewer's comments, it appears there may be some misunderstanding regarding our experimental setup and evaluation protocol. The stated weakness concerning performance gains does not reflect our main quantitative results. We fully respect the reviewer's perspective, but we kindly ask that this context be taken into consideration during the final decision process, especially as we were unable to engage in further clarification due to the system issue.
>
> **Reviewer b67S:** We clarified the role of the reference view and robustness when all inputs are inconsistent (**[Fig. 15, 16]**), provided additional high-resolution qualitative and quantitative results demonstrating the absence of block artifacts due to our transformer-based bilateral grid design (**[Fig. 7, 8, 9]**), detailed our training data scale (**[Lines 317, 320]**), and further justified the 3D foundation model-based self-supervised loss and its dependence on AnySplat via ablations (**[Tab. 3, Lines 309-311, Lines 503-507, Fig. 17, 19]**).
>
> **Reviewer bJY1:** We clarified that the reported runtimes indeed include CHROMA inference, which amounts to only a few seconds per scene on the same GPU setup (**[Lines 371-373]**), not the 0.04s stated by the reviewer. We explicitly acknowledged in the Limitations that our method requires pre-training a network (**[Lines 524-528]**). We provided a quantitative comparison between our bilateral grid head and a dense prediction head (**[Tab. 8, Lines 892-908]**), showing that the bilateral grid yields clearly better reconstruction performance while remaining resolution-flexible.
>
> -----------------------------------------------------------------------
>
> We hope that this summary helps contextualize how we addressed the reviewers' concerns and clarifies the extent of our revisions and discussion-phase contributions. Please let us know if any additional information or clarification would be helpful. Finally, we sincerely hope that our rebuttal responses will be taken into careful consideration when making the final decision.

---

### Meta-Review · Area_Chair_XPLC · 2026-01-07

**Summary:**

The reviewers generally found the method technically sound, well written, and empirically validated, with meaningful practical value and strong results. However, several concerns were raised:

Reviewer u8mx noted limited discussion/analysis of challenging cases (e.g., specularities), unclear qualitative gains in some examples, and missing visualization/analysis of confidence maps and dataset/model-assumption effects.

Reviewer UcJY expressed concern that training and testing under similar handcrafted ISP variations may inflate performance significance, and that gains over some baselines are modest.

Reviewer b67S pointed out reliance on a “good” reference view, potential block artifacts at higher resolution, and limited benefit relative to added training cost for the 3D-FM supervision.

Reviewer bJY1 viewed the work as an incremental but solid contribution, also questioning whether the reported efficiency numbers fully reflect processing cost and noting the need to mention network-training overhead as a limitation.

Despite these concerns, the paper’s practicality, clarity, and overall performance motivated the suggested acceptance.

**Reviewer Concerns:**

It appears that the reviewers' concerns have been properly addressed.

In particular, reviewer UcJY made a valid point that training and testing under similar handcrafted ISP variations does not convincingly demonstrate the model’s generalizability. In their rebuttal, the authors noted that they also evaluated the method on two additional public datasets, which provides some empirical evidence supporting generalizability.

**Reviewer Scores:**

Reviewer u8mx: 6
Reviewer UcJY: 4
Reviewer b67S: 6
Reviewer bJY1: 8

I expect that reviewers u8mx, b67S, and bJY1 will maintain their original scores, while reviewer UcJY is likely to increase the score to 5 or 6.

---

### Decision · Program_Chairs · 2026-01-26

Accept (Poster)